# Placental uptake and metabolism of 25(OH)vitamin D determine its activity within the fetoplacental unit

**Brogan Ashley[1†], Claire Simner[1†], Antigoni Manousopoulou[2,3], Carl Jenkinson[4], Felicity Hey[5,6], Jennifer M Frost[7], Faisal I Rezwan[1,8], Cory H White[1,9], Emma M Lofthouse[1], Emily Hyde[1], Laura DF Cooke[1], Sheila Barton[10], Pamela Mahon[10], Elizabeth M Curtis[10], Rebecca J Moon[10], Sarah R Crozier[10,11], Hazel M Inskip[10], Keith M Godfrey[10,12], John W Holloway[1], Cyrus Cooper[10,12,13], Kerry S Jones[5,6], Rohan M Lewis[1], Martin Hewison[4], Spiros DD Garbis[3], Miguel R Branco[7], Nicholas C Harvey[10,12‡], Jane K Cleal[1*‡]**

[1]The Institute of Developmental Sciences, Human Development and Health, Faculty of Medicine University of Southampton, Southampton, United Kingdom; [2]Beckman Research Institute, City of Hope National Medical Center, Duarte, United States; [3]Proteas Bioanalytics Inc, BioLabs at the Lundquist Institute, Torrance, United States; [4]Institute of Metabolism and Systems Research, The University of Birmingham, Birmingham, United Kingdom; [5]NIHR Cambridge Biomedical Research Centre, Nutritional Biomarker Laboratory. MRC Epidemiology Unit, University of Cambridge School of Clinical Medicine, Clifford Allbutt Building, Cambridge Biomedical Campus, Cambridge, United Kingdom; [6]Formerly at MRC Elsie Widdowson Laboratory, Cambridge, CB1 9NL l Merck Exploratory Science Center, Merck Research Laboratories, Cambridge, United States; [7]Centre for Genomics and Child Health, Blizard Institute, Barts and The London School of Medicine and Dentistry, Queen Mary University of London, London, United Kingdom; [8]School of Water, Energy and Environment, Cranfield University, Cranfield, United Kingdom; [9]Merck Exploratory Science Center, Merck Research Laboratories, Cambridge, United States; [10]MRC Lifecourse Epidemiology Centre, University of Southampton, Southampton, United Kingdom; [11]NIHR Applied Research Collaboration Wessex, Southampton Science Park, Southampton, United Kingdom; [12]NIHR Southampton Biomedical Research Centre, University of Southampton and University Hospital Southampton NHS Foundation Trust, Southampton, United Kingdom; [13]NIHR Oxford Biomedical Research Center, University of Oxford, Oxford, United Kingdom

*For correspondence:
j.k.cleal@soton.ac.uk

†These authors contributed equally to this work
‡These authors also contributed equally to this work

**Abstract** Pregnancy 25-hydroxyvitamin D [25(OH)D] concentrations are associated with maternal and fetal health outcomes. Using physiological human placental perfusion and villous explants, we investigate the role of the placenta in regulating the relationships between maternal 25(OH)D and fetal physiology. We demonstrate active placental uptake of 25(OH)D$_3$ by endocytosis, placental metabolism of 25(OH)D$_3$ into 24,25-dihydroxyvitamin D$_3$ and active 1,25-dihydroxyvitamin D [1,25(OH)$_2$D$_3$], with subsequent release of these metabolites into both the maternal and fetal circulations. Active placental transport of 25(OH)D$_3$ and synthesis of 1,25(OH)$_2$D$_3$ demonstrate that fetal supply is dependent on placental function rather than simply the availability of maternal 25(OH)D$_3$. We demonstrate that 25(OH)D$_3$ exposure induces rapid effects on the placental transcriptome and proteome. These map to multiple pathways central to placental function and thereby fetal

development, independent of vitamin D transfer. Our data suggest that the underlying epigenetic landscape helps dictate the transcriptional response to vitamin D treatment. This is the first quantitative study demonstrating vitamin D transfer and metabolism by the human placenta, with widespread effects on the placenta itself. These data demonstrate a complex interplay between vitamin D and the placenta and will inform future interventions using vitamin D to support fetal development and maternal adaptations to pregnancy.

## Editor's evaluation

The importance of vitamin D and its metabolism to maternal and fetal outcomes is well recognized. Both mother and fetus (in terms of skeletal development and birth weight) are heavily dependent on sufficient 25(OH) vitamin D and its conversion to 1,25(OH)$_2$ vitamin D. This interesting work concentrates on the central role of placental function and not maternal physiology to the uptake of 25(OH) vitamin D and its transformation to 1,25(OH)$_2$ vitamin D in determining the availability of vitamin D metabolites to the fetus in pregnancy. This work extends substantially our understanding of fundamental human fetal developmental processes.

## Introduction

Vitamin D (calciferol) cannot be synthesized by the fetus, so maternal vitamin D or its biologically significant metabolites 25-hydroxyvitamin D [25(OH)D] and/or 1,25-dihydroxyvitamin D [1,25(OH)$_2$D] must be transferred across the placenta. This transfer is important for both fetal and lifelong health. Indeed, maternal 25(OH)D concentrations are positively associated with fetal bone growth and birth weight (*Harvey et al., 2014c*; *Mahon et al., 2010*), and associations with bone health and body composition persist into postnatal life (*Crozier et al., 2012*; *Javaid et al., 2006*; *Cooper et al., 2016*; *Harvey et al., 2014a*; *Boyle et al., 2017*). Therefore, understanding the regulatory mechanisms and rate-limiting steps of placental vitamin D transfer is a prerequisite for identifying options for targeted intervention to improve pregnancy outcomes. It is unclear whether the benefits of vitamin D and its metabolites are due to direct transfer to the fetus or indirect effects on the placenta. Fundamental questions need to be answered: (1) Does placental metabolism generate substantial quantities of vitamin D metabolites that contribute to the maternal or fetal circulation? (2) How are these vitamin D metabolites taken up by placenta? (3) Do vitamin D metabolites affect the placenta itself? This third question is particularly important as changes in placental function could subsequently influence the fetus independently of vitamin D transfer.

Maternal 25(OH)D was previously thought to diffuse passively across the placenta, and to be hydroxylated in the fetus to 1,25(OH)$_2$D, as fetal metabolite levels relate to maternal levels for 25(OH)D (*Bouillon et al., 1981*; *Delvin et al., 1982*; *Markestad et al., 1984*) but not for 1,25(OH)$_2$D (*Fleischman et al., 1980*; *Hollis and Pittard, 1984*). We challenge this idea as work in the kidney questions the role of passive diffusion as a mechanism of vitamin D uptake and demonstrates that renal vitamin D uptake is driven primarily by receptor-mediated endocytosis of vitamin D bound to vitamin D-binding protein (DBP) (*White and Cooke, 2000*) or albumin (*Bikle et al., 1985*; *Bikle et al., 1986*). We propose a similar endocytic uptake mechanism for vitamin D [25(OH)D and 1,25(OH)$_2$D] in the placenta, indicating an active role for the placenta in regulating fetal 25(OH)D delivery. Furthermore, placental metabolism of maternal 25(OH)D, either by placental 1α-hydroxylase (*CYP27B1*) metabolism into 1,25(OH)$_2$D or 24-hydroxylase (*CYP24A1*)-mediated breakdown, will influence the quantity and form of vitamin D metabolite reaching the fetus. In human placenta, both enzymes are localized to the syncytiotrophoblast, which is the primary barrier to maternal-fetal exchange. Whether the activity of these placental enzymes generates substantial quantities of vitamin D metabolites that significantly contribute to the maternal or fetal circulations is unknown.

Placental uptake and metabolism of 25(OH)D and 1,25(OH)$_2$D will determine the availability of 1,25(OH)$_2$D for inducing transcriptional changes within the placenta via the vitamin D receptor (VDR), retinoid X receptor-alpha (RXRA) receptor dimer. Placental 1,25(OH)$_2$D may therefore regulate gene pathways involved in placental functions that influence fetal growth and risk of lifelong poor health. Indeed, we have shown that maternal 25(OH)D and DBP concentrations relate to expression of placental genes, including those associated with fetal and postnatal lean mass (*Cleal et al., 2015*;

*Cleal et al., 2011*). The effects of vitamin D, thought to be via the 1,25(OH)$_2$D form, on placental gene expression are not clear as these are cell/tissue specific and depend on the chromatin state and DNA-binding proteins available (*Zeitelhofer et al., 2017*; *Fetahu et al., 2014*). Furthermore, vitamin D may induce epigenetic regulation of gene expression. For example, in immune cells ligand-bound VDR exerts epigenetic effects via interaction with chromatin modifiers and coactivator or corepressor proteins, such as histone modifiers (*Zeitelhofer et al., 2017*; *Fetahu et al., 2014*). Changes to DNA methylation have also been reported with vitamin D cellular incubation, patient supplementation (*Fetahu et al., 2014*; *Zhou et al., 2014*; *Curtis et al., 2019*), or in association with vitamin D status (*Harvey et al., 2014b*), supporting a role of VDR in altering DNA methylation.

The current study was designed to establish how 25(OH)D$_3$ is taken up, metabolized, and mediates gene expression within the human placenta using the more physiological perfusion model and intact villous explant culture systems, contrasting with the cell model approach of previous studies. We show novel active mechanisms of 25(OH)D$_3$ uptake by the human placenta and placental metabolism of 25(OH)D$_3$ influencing fetal and maternal levels. Furthermore, 25(OH)D induces placental-specific effects on the transcriptome, the native and in vivo modified proteome expressed in patterns relevant to placental function and therefore fetal development, independent of vitamin D transfer. These effects are dependent on the underlying epigenetic landscape. By discovering the mechanisms underlying the complex interactions between maternal vitamin D levels, placental vitamin D handling, and genetic/epigenetic processes in the placenta, our findings inform the identification of functional biomarkers that relate to fetal growth and the risk of disease across the life course.

## Results and discussion

### 25(OH)D$_3$ is taken up by the human placenta and both metabolized and transferred into the fetal circulation

The use of $^{13}$C-labeled 25(OH)D$_3$ in the maternal circulation of the ex vivo placental perfusion model allowed actual quantification of the placental metabolism of $^{13}$C-25(OH)D$_3$ and transfer of this $^{13}$C-25(OH)D and its metabolites from the maternal to fetal circulation in the term placenta. Although in vivo maternal and fetal circulating 25(OH)D$_3$ levels at term may correlate (*Park et al., 2017*), the level of actual 25(OH)D transfer and metabolism by the human placenta is unclear as prior data provided inconsistent results using in vivo plasma measures or closed loop perfusion experiments (*Ron et al., 1984*). We investigated the metabolism of $^{13}$C-25(OH)D$_3$ within placental tissue and transfer across the placenta using the whole placental cotelydon perfusion open loop methodology as adapted in our laboratory (*Cleal et al., 2011*). Using five fresh placentas, $^{13}$C-25(OH)D$_3$ (plus albumin as a binding protein) was perfused for 5 hr into the maternal circulation of a single isolated cotyledon and the amount transferred into the placenta and fetal circulation quantified using liquid chromatography hyphenated with mass spectrometry (LC-MS/MS). The amount of each specific vitamin D metabolite transferred into the fetal and maternal circulation was also quantified using LC-MS/MS or enzyme immunoassay (EIA) (*Figure 1—source data 1*).

Following 5 hr of placental perfusion, there was loss of maternal $^{13}$C-25(OH)D$_3$ from the maternal circulation indicative of placental uptake (27.4% ± 6.3% of maternal stock) and release into the fetal circulation (5.1% ± 1.5% of maternal stock; *Figure 1a*). These amounts are consistent with the levels taken up by the kidney via active mechanisms (*Nykjaer et al., 1999*). The transfer rate of $^{13}$C-25(OH)D$_3$ into the placenta was 3.01 ± 1.12 pmol/min/g, with 0.45 ± 0.27 pmol/min/g stored in placental tissue, 2.25 ± 0.97 pmol/min/g metabolized, and 0.30 ± 0.10 pmol/min/g transferred to the fetal circulation (*Figure 1b*). However, there was no correlation between $^{13}$C-25(OH)D$_3$ levels in the maternal and fetal circulations of this open loop placental perfusion system: a system that identifies delivery, but not build-up, of substrates in either circulation (*Figure 1g*). Together, these data suggest that the placenta plays an important role in determining the amount of 25(OH)D$_3$ transferred to the fetus and also stores and uses 25(OH)D$_3$ for its own cellular needs. Placental metabolism and uptake of $^{13}$C-25(OH)D$_3$ are therefore key factors in determining fetal $^{13}$C-25(OH)D$_3$ supply (*Figure 2a*).

The primary barrier to transfer, the placental syncytiotrophoblast, expresses the vitamin D-metabolizing enzymes 24-hydroxylase (*CYP24A1*) and 1α-hydroxylase (*CYP27B1*) (*Simner et al., 2020*). There was direct evidence of 24-hydroxylase action in the placental tissue with metabolism of 25(OH)D$_3$ into $^{13}$C-24,25(OH)$_2$D$_3$ (*Figure 1c and d*) but no detectable epi25(OH)D$_3$.

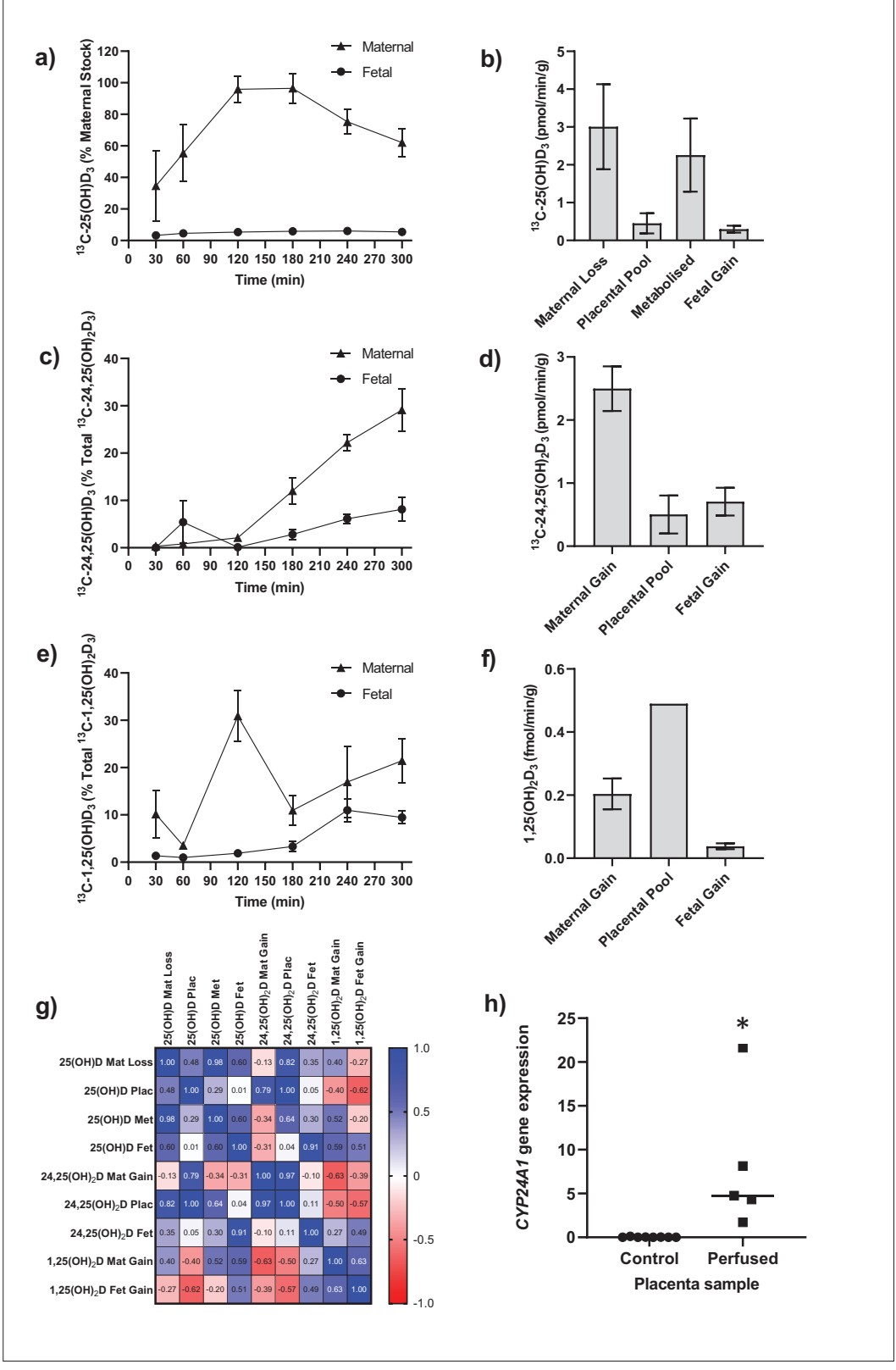

**Figure 1.** Transfer and metabolism of $^{13}C$-25(OH)D$_3$ by the term perfused human placenta over 5 hr. (**a**) $^{13}C$-25(OH)D$_3$ in maternal and fetal circulations as a % of maternal perfusate concentration. (**b**) Rate of maternal $^{13}C$-25(OH)D$_3$ lost from the maternal circulation, accumulated in placental tissue, metabolized, or transferred to the fetal circulation. (**c**) $^{13}C$-24,25(OH)$_2$D$_3$ transfer into the maternal and fetal circulations as a % of $^{13}C$-24,25(OH)$_2$D$_3$

*Figure 1 continued on next page*

*Figure 1 continued*

metabolized by the placenta. (**d**) Rate of placental production of $^{13}$C-25(OH)D$_3$ and transfer rate into the maternal and fetal circulations. (**e**) 1,25(OH)$_2$D$_3$ transfer into the maternal and fetal circulations as a % of 1,25(OH)$_2$D$_3$ produced by the placenta. (**f**) Rate of placental production of $^{13}$C-1,25(OH)$_2$D$_3$ and 1,25(OH)$_2$D$_3$ transfer into the maternal and fetal circulations. (**g**) Pearson's correlations between metabolites. (**h**) Placental *CYP24A1* relative mRNA expression was significantly increased in $^{13}$C-25(OH)D$_3$ perfused placental samples compared to non-perfused control placental tissue samples (*p<0.05). Data are presented as mean (SEM).

The online version of this article includes the following source data for figure 1:

**Source data 1.** Metabolism data.

Placental $^{13}$C-25(OH)D$_3$ and $^{13}$C-24,25(OH)$_2$D levels were positively correlated ($r$ = 0.998, p=0.0001; *Figure 1g*). Of this $^{13}$C-24,25(OH)$_2$D$_3$, 16.78% ± 7.20% (0.50 ± 0.30 pmol/min/g) was stored in the placental tissue, 20.4% ± 3.21% (0.71 ± 0.10 pmol/min/g) was transferred into the fetal circulation, and 66.56% ± 6.31% (2.05 ± 0.35 pmol/min/g) was transferred into the maternal circulation (*Figure 1c and d*). There was also evidence of placental 1α-hydroxylase activity with metabolism of 25(OH)D$_3$ into 1,25(OH)$_2$D$_3$ (*Figure 1e and f*). Transfer of 1,25(OH)$_2$D$_3$ into both the maternal (0.20 ± 0.05 fmol/min/g) and fetal (0.04 ± 0.01 fmol/min/g) circulations occurred over the 5 hr experiment with direct evidence of placental metabolism of $^{13}$C-25(OH)D$_3$ into $^{13}$C-1,25(OH)$_2$D$_3$ (0.49 fmol/min/g, n = 1 due to detection limits). These data provide evidence that once within the placenta 25(OH)D can be metabolized into downstream products that could have effects within the placenta itself including VDR-mediated transcription. These findings demonstrate that the human placenta does produce 1,25(OH)$_2$D$_3$ and release it into the maternal circulation at levels anticipated to increase overall circulating concentrations of this metabolite in the mother. However, the placental contribution is likely to be a smaller proportion of overall 1,25(OH)$_2$D$_3$ concentration in the maternal circulation as most of this metabolite is produced by the maternal kidney. Case

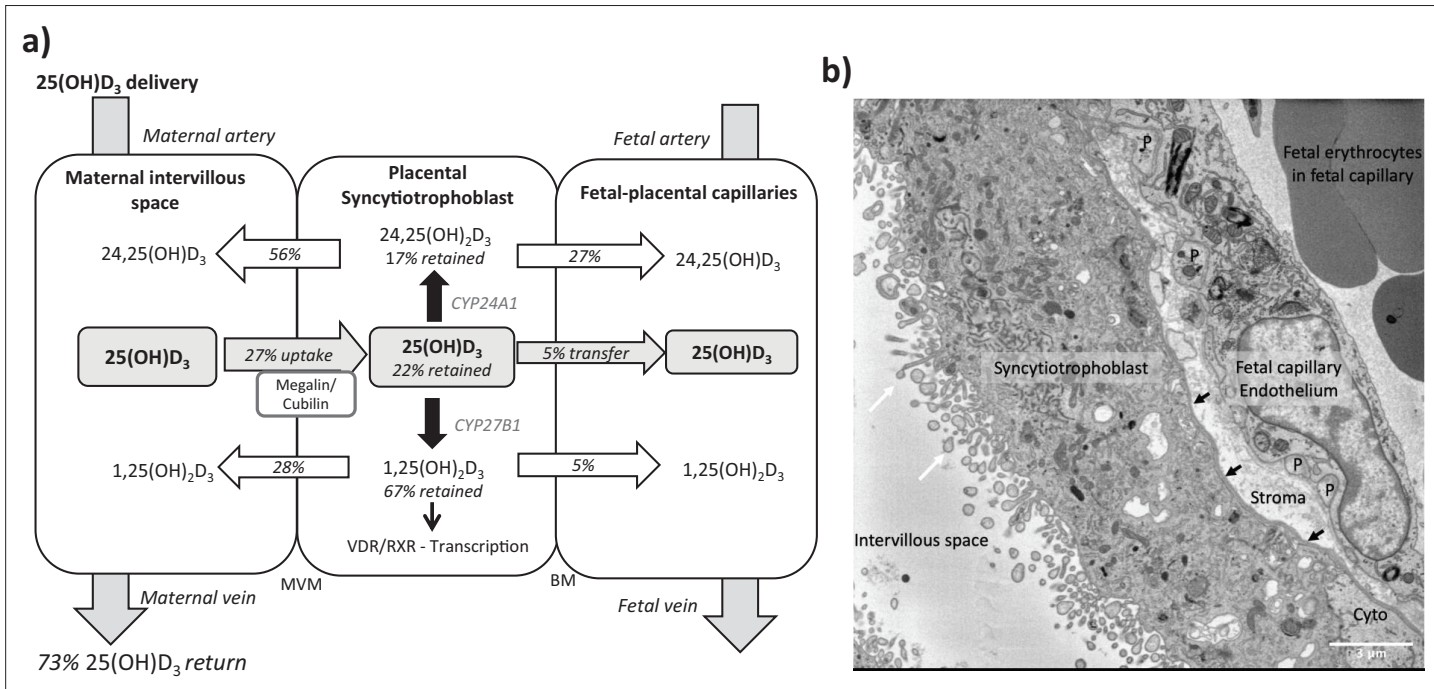

**Figure 2.** Summary of vitamin D transport and metabolism by the placenta. (**a**) Schematic model of $^{13}$C-25(OH)D$_3$ transfer and metabolism by the term perfused human placenta over 5 hr. (**b**) Electron microscopy image showing a cross section of the human placental barrier at term. The intervillous space is filled with maternal blood, the syncytiotrophoblast forms a continuous barrier across the surface of the villi, the microvilli on the apical plasma membrane are indicated by white arrows, and the syncytiotrophoblast basal membrane can be seen abutting the trophoblast basal lamina, which is indicated by black arrows. A small region of cytotrophoblast can be seen labeled 'cyto' between the syncytiotrophoblast and trophoblast basal lamina. The connective tissue of the villous stroma lies between the trophoblast and the fetal capillaries. The stroma also contains fibroblasts and macrophages that are not shown here. Pericyte fingers around the fetal capillary are labeled 'P'. The fetal capillary endothelial cells form the fetal blood vessel.

reports of pregnant women with impaired renal function showed that circulating $1,25(OH)_2D$ levels increased during pregnancy but at levels much lower than normal pregnant women (*Greer et al., 1984*; *Turner et al., 1988*). Furthermore, high levels of $1,25(OH)_2D$ (uncoupled to DBP levels) are observed throughout pregnancy without hypercalciuria or hypercalcemia. This could be of placental origin, but is most likely via renal 1α-hydroxylase that is uncoupled from feedback control (*Hollis et al., 2011*).

We observed greater release of $^{13}C$-$25(OH)D_3$ metabolites into the maternal circulation compared with the fetal circulation (*Figure 2a*). The transfer of $^{13}C$-$25(OH)D_3$ and its metabolites in the placenta to fetal direction may be more challenging, and in particular, diffusion across the water-filled villous stroma may prove a barrier to their diffusion (*Figure 2b*), consistent with recent observations on placental cortisol transfer (*Stirrat et al., 2018*). Indeed, positive correlations between fetal $^{13}C$-$25(OH)D_3$ and $^{13}C$-$24,25(OH)_2D$ concentrations ($r = 0.913$, p=0.03; *Figure 1g*) suggest similar or limited transfer processes for these metabolites in the placental to fetal direction. This is supported by the fact that placental to maternal transfer rates were higher, prioritizing maternal supply, and that placental and maternal $^{13}C$-$24,25(OH)_2D$ levels positively correlated ($r = 0.966$, p=0.03; *Figure 1g*). The greater amount of $^{13}C$-$25(OH)D_3$ metabolites released into the maternal circulation, compared with the fetal circulation, also indicates a potential role of the placenta in vitamin D-mediated maternal adaptations to pregnancy. These data substantiate a role for placental metabolism and transfer mechanisms being important for fetal vitamin D metabolite levels. These experiments therefore show that the placenta can produce substantial quantities of $25(OH)D_3$ metabolites, which can contribute to both the maternal and fetal circulating levels and highlights the importance of placental expression of the vitamin D-metabolizing enzymes 24-hydroxylase (gene *CYP24A1*) and 1α-hydroxylase (gene *CYP27B1*).

Placental *CYP24A1* gene expression (which is induced by $1,25(OH)_2D_3$) was significantly increased following perfusion with $^{13}C$-$25(OH)D_3$ compared with control (*Figure 1h*), providing further evidence that $25(OH)D_3$ is converted to active $1,25(OH)_2D_3$ by 1α-hydroxylase in human placenta. In a subset of term human placentas taken from the Southampton Women's Survey (SWS, n = 102), we did not observe an association between maternal 34-week pregnancy plasma $25(OH)D_3$ levels and *CYP24A1* ($r = -0.15$, p=0.18) or *CYP27B1* ($r = 0.14$, p=0.20) gene expression (*Cleal et al., 2015*; *Inskip et al., 2006*). In the SWS placentas, *CYP24A1* mRNA expression was associated with maternal mid-upper arm muscle area pre-pregnancy ($r = 0.33$, p=0.001 n = 101), at 11 weeks' gestation ($r = 0.33$, p=0.02, n = 76) and at 34 weeks' gestation ($r = 0.26$, p=0.01, n = 95). This suggests that factors other than vitamin D levels, such as maternal body composition, may regulate baseline levels of this gene. The fact that the placenta can produce $1,25(OH)_2D_3$, which induces expression of the degradation enzyme *CYP24A1*, suggests homeostatic regulation of placental vitamin D activity and potentially its transfer to the fetus. This has implications for the effectiveness of vitamin D supplementation during pregnancy.

## Uptake of $25(OH)D_3$ and albumin into the placenta is prevented by inhibition of endocytosis

To explore the mechanisms of placental $25(OH)D_3$ uptake, fresh term human placental villous fragments were incubated for 8 hr with 20 µM $25(OH)D_3$ plus the binding protein albumin. Analysis by qPCR revealed a significant increase in placental *CYP24A1* gene expression in villous fragments following incubation with $25(OH)D_3$ compared to control (*Figure 3*, *Figure 3—source data 1*). In the presence of albumin, *CYP24A1* mRNA expression was increased compared to $25(OH)D_3$ alone, suggesting that albumin may facilitate $25(OH)D_3$ uptake (*Figure 3a*). We suggest that receptor-mediated endocytosis may play an important role in the uptake of vitamin D into the human placenta as seen in the kidney. The increased *CYP24A1* expression with vitamin D uptake is increased further by the presence of a binding protein; this could be due to increased solubility or receptor-mediated binding for the uptake mechanism.

The placental uptake mechanisms for $25(OH)D_3$ via the carrier protein albumin were investigated by measuring uptake of FITC-labeled albumin into placental villous fragments. Uptake over 30 min at 4°C (a temperature that blocks endocytosis) and 37°C showed a significant effect of both time and temperature on placental FITC-albumin uptake, indicating an active uptake mechanism (*Figure 3b and c*, *Figure 3—source data 1*). In addition, FITC-dextran uptake was not observed, suggesting that the albumin uptake mechanism is selective (data not shown).

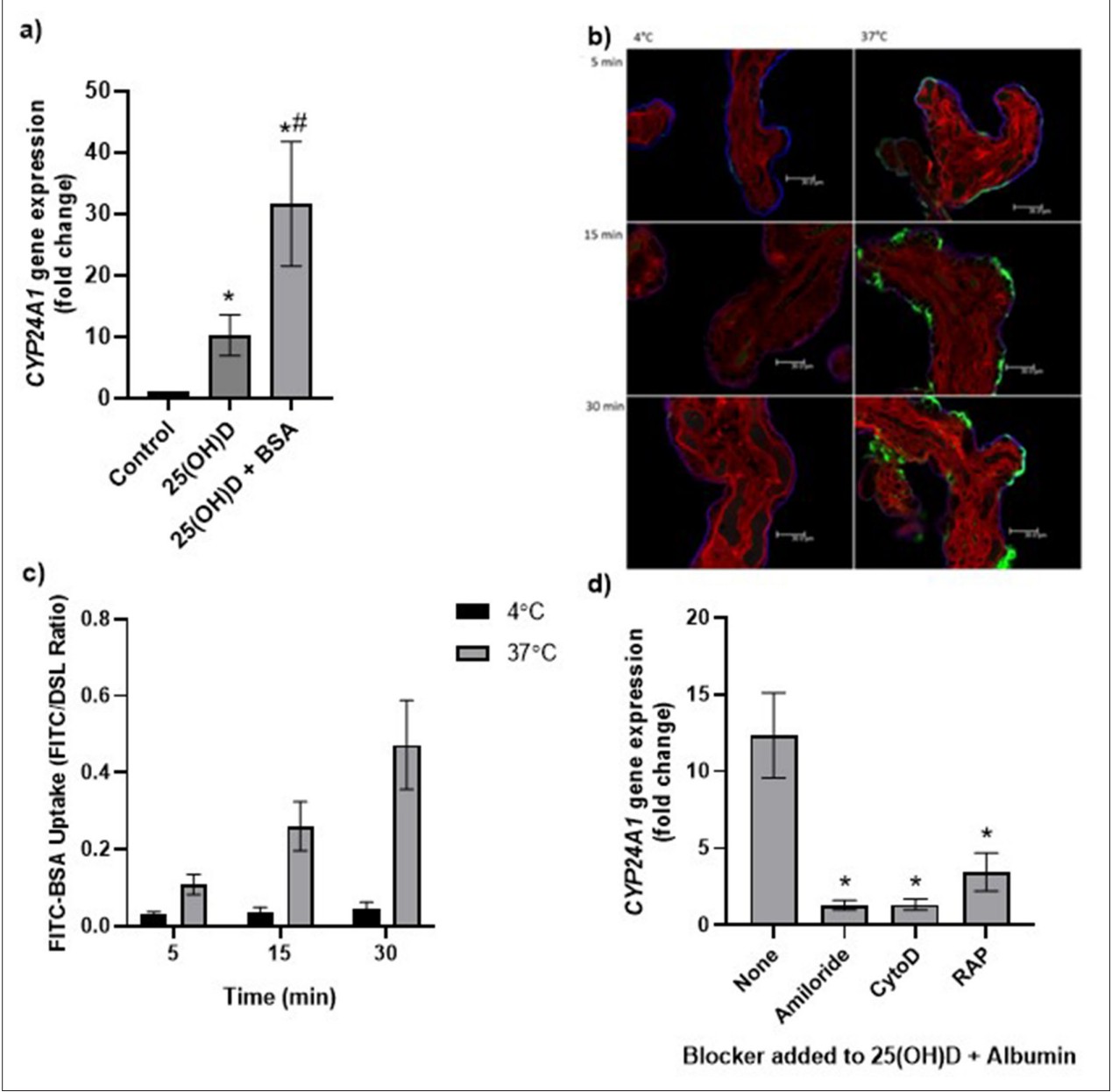

**Figure 3.** Uptake of 25(OH)D₃ in placental villous fragments is facilitated by albumin and mediated by endocytic processes. (**a**) Relative *CYP24A1* mRNA expression was increased in placental villous fragments incubated with 25(OH)D₃ (n = 15) compared to control (ethanol plus albumin, n = 15; *p<0.001) and was increased further with 25(OH)D₃ and albumin (n = 6) compared to 25(OH)D₃ alone (#p<0.05). Uptake of FITC-albumin to placental fragments is mediated by endocytic mechanisms. (**b**) Representative images showing FITC-albumin uptake into placental villous fragments at 5, 15, and 30 min. Green, FITC-albumin; red, villous stroma stained by rhodamine-PSA; blue, MVM stained by biotin-DSL. (**c**) FITC-albumin uptake increased with time (p=0.03) and temperature (p=0.03), n = 3. (**d**) *CYP24A1* gene expression was reduced by addition of the blockers amiloride, cytochalasin D (CytoD), and receptor-associated protein (RAP) compared to 25(OH)D plus albumin-stimulated expression with no blocker (*p<0.05; n = 4–5). BSA, bovine serum albumin. Data are presented as mean (SEM).

The online version of this article includes the following source data for figure 3:

**Source data 1.** Uptake data.

We next investigated whether binding protein-mediated vitamin D uptake occurred via endocytosis. In the kidney, both DBP and albumin bind to the plasma membrane receptors megalin (LDL-related protein 2 [*LRP2*]) and cubilin (*CUBN*) (*Birn et al., 2000*; *Doherty and McMahon, 2009*), which mediate vitamin D internalization by clathrin-dependent endocytosis (*Nykjaer et al., 1999*; *Nykjaer et al., 2001*). Both receptors are expressed in the primary barrier to transfer the syncytiotrophoblast,

suggesting a role for vitamin D uptake by the placenta (*Simner et al., 2020*). To investigate this, fresh placental villous fragments (n = 4–5 placentas per treatment) were incubated with 20 µM 25(OH)D$_3$ and albumin, plus the endocytic inhibitors 5 mM amiloride, 10 µM cytochalasin D, and 2 µM receptor-associated protein (RAP), which block pinocytosis, classical clathrin-dependent endocytosis, and megalin-mediated endocytosis (one type of clathrin-dependent endocytosis), respectively. The application of a range of endocytic inhibitors, which target different aspects of endocytic uptake, aimed to pinpoint the discrete machinery required for the uptake of this particular cargo (*Cooke et al., 2021*). The vitamin D-mediated upregulation of *CYP24A1* mRNA expression was significantly reduced by amiloride, cytochalasin D, and RAP in the presence of albumin (*Figure 3d*, *Figure 3—source data 1*). Although the Na$^+$/H$^+$ exchange inhibitor amiloride was used in our study to inhibit pinocytosis, it has been shown to also inhibit clathrin-mediated uptake in epithelial cells (*Ivanov et al., 2004*) and receptor-mediated uptake in kidney cells (*Gekle et al., 2001*), suggesting that our observed inhibitory effects of amiloride may be due to nonselective blockage of endocytosis via modulation of the actin cytoskeleton. The inhibitory effect of cytochalasin D suggests that 25(OH)D$_3$ is taken up by the human placenta using a clathrin-dependent endocytic mechanism, and specifically blocking this mechanism by RAP indicates a role for binding protein-mediated uptake via specific megalin receptor binding. These observations suggest that vitamin D is transported into the placenta by a selective and controlled active mechanism, not by simple diffusion as previously thought, which has potential implications for the effectiveness of maternal vitamin D supplementation and its bioavailability to the fetus.

The gene expression of megalin/*LRP2* and cubilin/*CUBN* was investigated in term human placentas from the SWS (*Inskip et al., 2006*). Both showed no association with maternal 34-week pregnancy plasma 25(OH)D$_3$ levels (data not shown) and positive associations with measures of fetal size; *LRP2*

**Table 1.** Associations of offspring size and vitamin D receptor relative mRNA expression in the Southampton Women's Survey.

| | VDR mRNA expression | | |
|---|---|---|---|
| | r | p | n |
| *Size z-scores* | | | |
| 19-week HC | −0.20 | 0.25 | 58 |
| 19-week AC | −0.03 | 0.83 | 58 |
| 19-week FL | −0.05 | 0.74 | 58 |
| 34-week HC | −0.33 | 0.001 | 59 |
| 34-week AC | −0.32 | 0.02 | 59 |
| 34-week FL | −0.28 | 0.04 | 59 |
| *Birth data* | | | |
| Placenta weight (g) | −0.22 | 0.03 | 101 |
| Birth weight (g) | −0.19 | 0.06 | 102 |
| CHL (cm) | −0.22 | 0.03 | 102 |
| DXA lean mass (g) | −0.24 | 0.02 | 102 |
| DXA fat mass (g) | −0.22 | 0.03 | 102 |
| *4-year data* | | | |
| DXA lean mass (kg) | −0.42 | 0.005 | 46 |
| DXA fat mass (kg) | −0.35 | 0.02 | 46 |
| Weight (kg) | −0.32 | 0.02 | 56 |

AC: abdominal circumference; CHL: crown-heel length; DXA: dual-energy X-ray absorptiometry measurements; FL: femur length; HC: head circumference; VDR: vitamin D receptor.

The online version of this article includes the following source data for table 1:

**Source data 1.** qPCR vitamin D receptor data.

mRNA expression related to crown-rump length at 11 weeks' gestation ($r = 0.23$, p=0.05, n = 51) and birth weight ($r = 0.20$, p=0.05, n = 102); *CUBN* mRNA expression related to birth head circumference ($r = 0.22$, p=0.03, n = 102) and crown-heel length ($r = 0.21$, p=0.04, n = 102). This supports a role for the placenta in mediating the amount of vitamin D transferred to the fetus, which will be determined by placental uptake capacity, which in turn effects fetal size. We suggest that the placenta may be playing an active role in mediating the supply of vitamin D to the fetus, thus contributing to its regulation and effects on development.

Vitamin D-induced gene expression changes in the placenta could also mediate fetal development by altering placental function. Placental uptake and metabolism of $25(OH)D_3$ and $1,25(OH)_2D_3$ will ultimately determine the amount of $1,25(OH)_2D_3$ available within the placenta for activation of the VDR and initiation of transcription. Interestingly, in SWS placentas we see associations between VDR mRNA expression and measures of fetal size, suggesting that vitamin D-mediated transcription is important for regulating aspects of placental function that affect fetal size. Placental *VDR* mRNA levels associate negatively with measures of fetal and neonatal size and body composition (*Table 1*, *Table 1—source data 1*), and *RXRA* mRNA levels associate negatively with 4-year lean mass ($r = -0.27$, p=0.005) and fat mass ($r = -0.34$, p=0.03). The placental levels of VDR and RXR may therefore determine the utilization of available $1,25(OH)_2D$ by the placenta via VDR binding. Increased VDR/RXR levels will result in activation of the *CYP24A1* gene encoding the enzyme that breaks down $25(OH)D$ and $1,25(OH)_2D$, reducing the quantities of the two metabolites available for transfer to the fetus for mediating effects on growth.

## Short-term $25(OH)D_3$ exposure leads to transcriptomic changes in human placenta

Our previous experiments show that conversion of $25(OH)D_3$ into $1,25(OH)_2D_3$ within the human placenta initiates gene transcription of a specific vitamin D-responsive gene, *CYP24A1*. To understand the genome-wide effects of vitamin D on the placenta, RNA-sequencing analysis was performed on villous fragments from four fresh term human placentas. From each placenta, one set of fragments (in triplicate) was incubated with Tyrode's buffer containing 20 µM $25(OH)D_3$ plus 0.7 mM bovine serum albumin (BSA (treatment group)) and one set of fragments (in triplicate) was incubated in Tyrode's buffer with 0.7 mM BSA (control group) for 8 hr.

Principal component analysis (PCA) demonstrated condition clustering, indicating an effect of treatment (*Figure 4a*). Placental fragments exposed to $25(OH)D_3$ displayed marked changes in gene expression compared to those in control buffer (*Figure 4b*). Using a 5% false discovery rate (FDR), we observed 493 genes to be differentially expressed between $25(OH)D_3$-treated compared to control placental fragments (358 increased and 135 decreased; *Supplementary file 1*) and those with a fold change of 2 or more are presented (*Figure 4c*). The mRNA expression of a selection of genes was replicated in these samples by qPCR (*Figure 4—figure supplement 1*). We compared the gene expression changes observed following short-term vitamin D exposure with those in a published dataset from human placenta (GSE41331), which looked at longer-term vitamin D response (24 hr). Whilst considerable differences were expected and observed, it was reassuring that genes found to be upregulated at 8 hr also tended to have increased expression in the 24 hr dataset, whereas downregulated genes were unchanged (*Figure 4—figure supplement 1*). This probably reflects the fact that upregulated genes are the likely direct targets of VDR. Differentially expressed genes in common between both placental datasets included *CYP24A1* and *SNX31* (*Figure 4—figure supplement 1*).

To identify the biological functions regulated by vitamin D in human placenta, we performed pathway analysis incorporating all differentially expressed genes using ToppGene with the ToppFun subprogram, and significance was adjusted using the Benjamini–Hochberg correction depicted by –log(B-H p-value) and a significance threshold of 1%. The pathway analysis revealed that genes from the pathways involved in cytokine binding and immune or inflammatory responses were downregulated following $25(OH)D_3$ treatment (*Figure 4d*). These findings fit with the role of vitamin D in regulating cellular growth and immune function in a protective manner (*Zeitelhofer et al., 2017*). Genes from gene pathway terms related to transcriptional regulation were upregulated following $25(OH)$$D_3$ treatment (*Figure 4d*). These included many different transcription factors (e.g., *REST*, *FOXO3*), suggesting that this early response to $25(OH)D_3$ treatment maybe to cascade the signal to other transcriptional regulatory networks. Furthermore, transcription factors known to interact with VDR, such

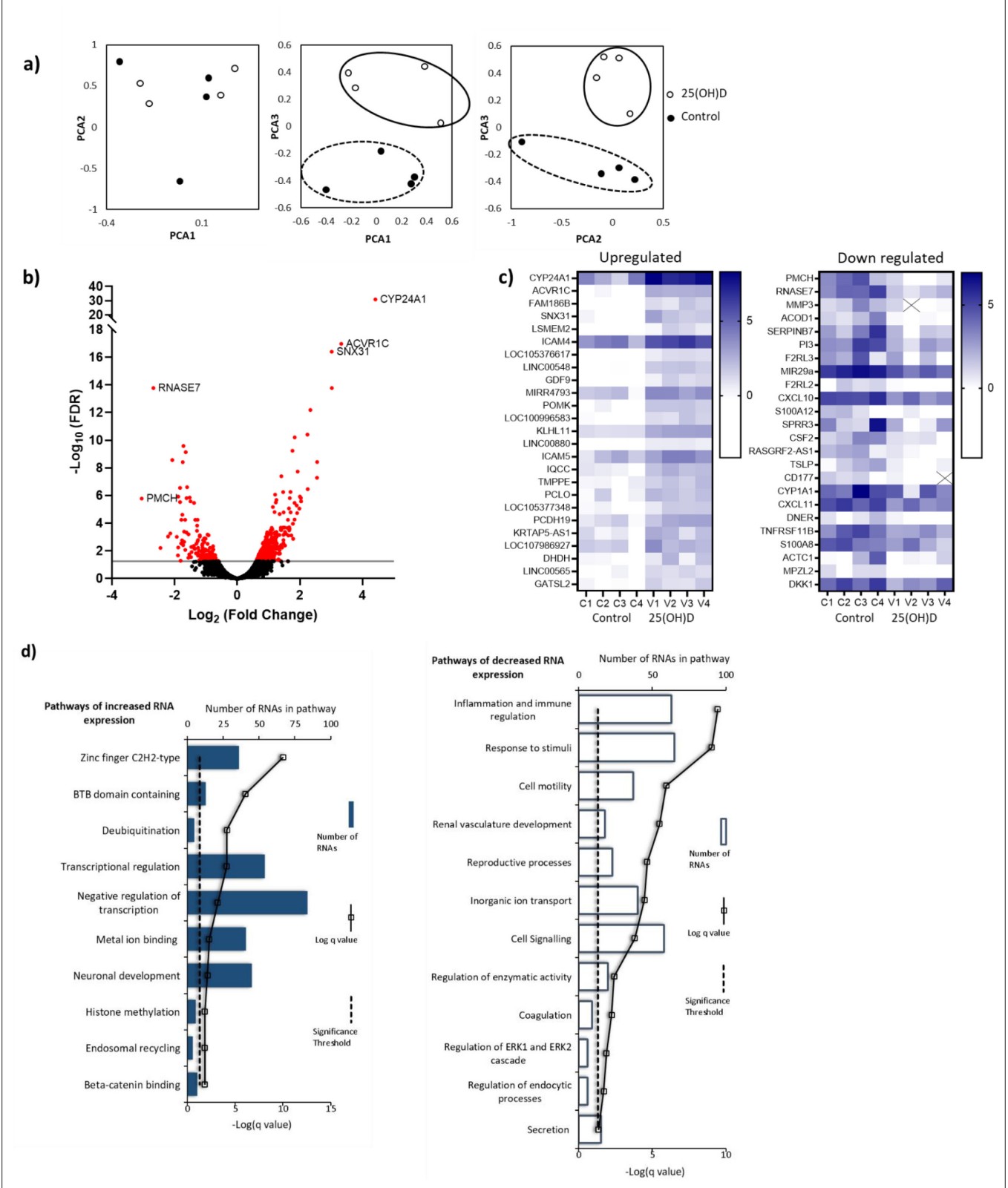

**Figure 4.** RNA-seq-derived gene signatures of human placental samples following 8 hr 25(OH)D₃ incubation. (**a**) Principal component analysis indicated clustering of 25(OH)D₃-treated samples (clear) compared with control samples (black). (**b**) Volcano plot showing 493 genes were differentially expressed (red) at a false discovery rate (FDR)-adjusted p-value<0.05 (gray line). (**c**) Differentially expressed genes with a fold change of 1.5 or above at FDR 0.05 are presented as heatmaps [log₂(normalized expression)]. (**d**) Pathway analysis (ToppGene) of all differentially expressed genes (no fold change cutoff)

*Figure 4 continued on next page*

Figure 4 continued

reveals both up- and downregulation of molecular function and biological process gene pathways following 8 hr 25(OH)D₃ incubation. For pathway analysis, significance was adjusted using the Benjamini–Hochberg correction depicted by –log(B&H q-value) with a significance threshold of 1% (dashed line).

The online version of this article includes the following figure supplement(s) for figure 4:

**Figure supplement 1.** Gene expression changes following vitamin D incubation.

as the central regulators of gene expression CREB-binding protein (*CREBBP*) and its paralog *EP300*, were upregulated, supporting the idea that the response to 25(OH)D₃ treatment is about potentiating a robust secondary response. Effects on key transcription factors such as *CREBBP* and *EP300,* which can also interact with enhancer-bound transcription factors to activate transcription and have histone acetyltransferase activity, indicate that epigenetic mechanisms may play a role in 25(OH)D₃-mediated regulation of gene expression.

## The underlying epigenetic landscape helps to dictate the transcriptional response upon vitamin D treatment

Short-term 25(OH)D₃ exposure leads to transcriptomic changes in human placenta; however, the role of epigenetic factors in this observation is unclear. We therefore investigated whether 8 hr incubation with Tyrode's buffer containing 20 µM 25(OH)D₃ plus 0.7 mM BSA (treatment group) altered DNA methylation in villous fragments (in triplicate) from eight fresh term human placenta compared to samples (in triplicate) from the same placenta treated with Tyrode's buffer with 0.7 mM BSA (control group). DNA methylation was measured using the Illumina EPIC 850K array, and CpGs with altered DNA methylation following 25(OH)D₃ treatment were identified using a Wilcoxon signed-rank test ($p < 0.05$).

Short-term 25(OH)D₃ exposure led to limited alterations in DNA methylation, with 319 CpGs displaying methylation differences larger than 10% (230 hypomethylated, 89 hypermethylated; *Supplementary file 2*). Most of these changes involved isolated CpGs, whereas clusters of two or more differentially methylated CpGs were only observed in two hypomethylated and six hypermethylated regions (*Figure 5a*). Whilst the methylation levels of the upregulated and downregulated genes identified in the RNA-seq data were not affected by 25(OH)D₃ treatment, their baseline levels differed. Notably, the promoters of the upregulated genes displayed markedly lower methylation than those of downregulated genes in both control and 25(OH)D₃-treated conditions (*Figure 5b*). Consistent with this observation, 72% of upregulated gene promoters overlapped with CpG islands, in contrast to 23% of the downregulated (and 56% genome-wide). These results suggest that the underlying epigenetic landscape helps to dictate the transcriptional response to 25(OH)D₃ treatment, possibly by enabling VDR binding at open chromatin regions.

To extend these observations, we performed ChIP-seq on syncytialized cytotrophoblast cells (*Desforges et al., 2015*) incubated with 20 µM 25(OH)D₃ or control cell culture medium for 24 hr (see Materials and methods; n = 2 placentas; *Figure 5c*). We profiled histone modifications associated with open chromatin, H3K4me3 and H3K27ac, the latter being also an in vivo target of EP300/CREBBP acetyltransferase activity. We detected no loci with significant changes in the enrichment of these marks when comparing control and vitamin D conditions (see Materials and methods for details). However, in line with the DNA methylation data, we found that the enrichment of these marks in the upregulated and downregulated genes identified in the RNA-seq data differed at baseline, but was not affected by 25(OH)D₃ treatment. Specifically, we found that promoters of upregulated genes displayed strikingly higher levels of both H3K4me3 and H3K27ac than those seen at downregulated genes in both control and 25(OH)D₃-treated conditions (*Figure 5d*). This pattern was also seen when focusing only on CpG island promoters (*Figure 5—figure supplement 1a*), ruling out the possibility that higher levels of H3K4me3/H3K27ac at upregulated genes were only due to increased CpG content, and thus reinforcing the idea that open chromatin poises genes for upregulation by VDR. Although our ChIP-seq data are from isolated cytotrophoblast, we observed very similar patterns in ENCODE data from 16-week placenta (*Figure 5—figure supplement 1b*).

Given these observations, we asked whether a preference for open chromatin was specific to the VDR transcriptional response in placenta. We therefore analyzed gene expression and open chromatin

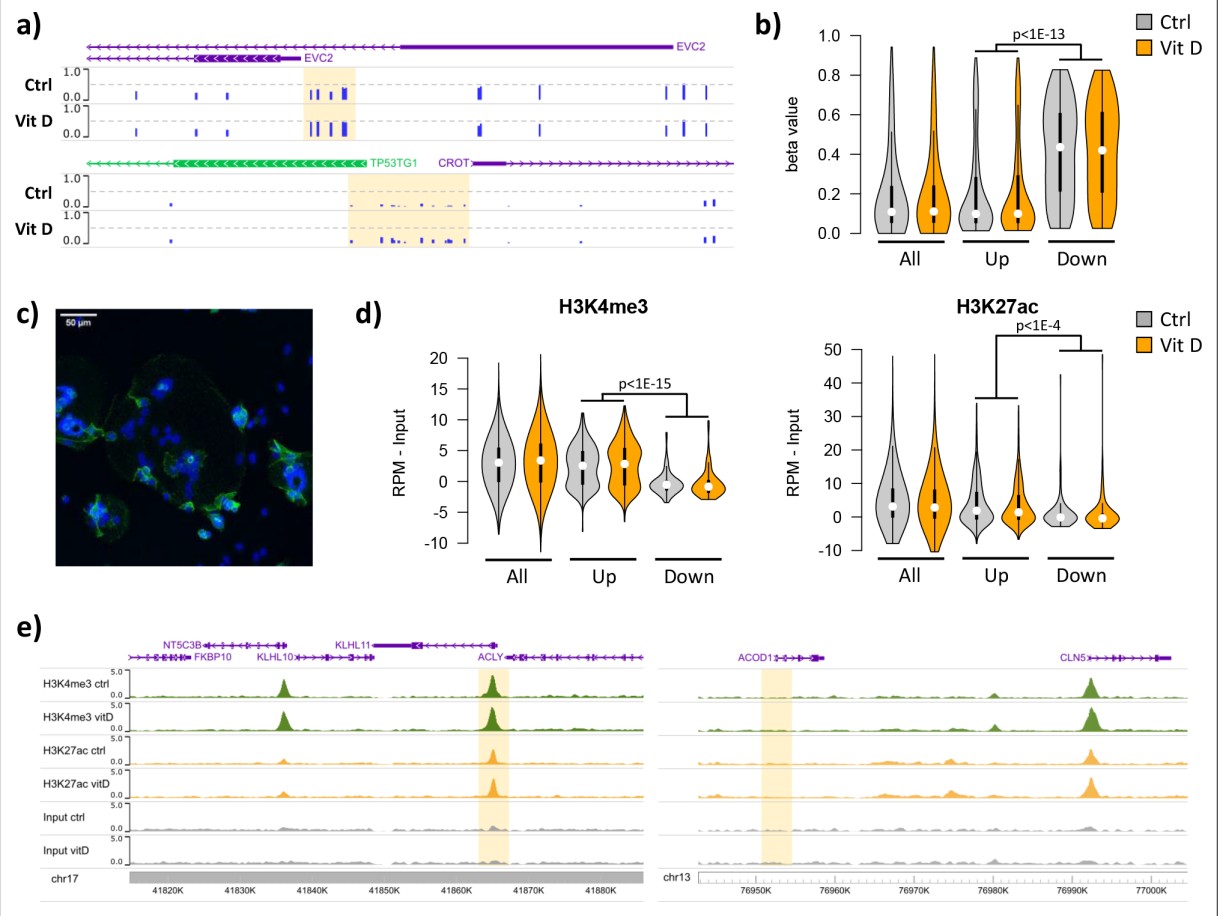

**Figure 5.** Short-term vitamin D exposure has limited effects on placental methylation, but the pre-existing epigenetic landscape has a major effect on vitamin D-mediated transcription. (**a**) Placental fragments were exposed for 8 hr to 25(OH)D$_3$, which led to limited alterations in DNA methylation compared to incubation with control buffer. Shown are two examples of clusters of hypermethylated CpGs (highlighted in yellow), where the blue bars represent the array's beta value for individual CpGs. (**b**) The promoters of the upregulated genes identified in the RNA-seq data displayed lower methylation than those of downregulated genes in both control and 25(OH)D$_3$-treated conditions. To extend these observations, we performed ChIP-seq on syncytialized cytotrophoblast cells incubated with 20 μM 25(OH)D$_3$ or control cell culture medium for 24 hr (n = 2 placentas). (**c**) Representative confocal microscopy image of cytotrophoblast cells cultured for 90 hr and stained with DAPI (blue; nuclei) and desmoplakin (green), present on the cell surface. Multiple nuclei within a single-cell demonstrate syncytialization has occurred. (**d**) The promoters of upregulated genes (identified in the RNA-seq data) displayed higher levels of both H3K4me3 and H3K27ac than those seen at downregulated genes. (**e**) Examples of specific upregulated (*KLHL11*) and downregulated (*ACOD1*) genes, showing no changes in the enrichment of H3K4me3 or H3K27ac at the promoter (highlighted in yellow) when comparing control and vitamin D conditions.

The online version of this article includes the following figure supplement(s) for figure 5:

**Figure supplement 1.** The effects of the pre-existing epigenetic landscape on vitamin D mediated transcription.

(FAIRE-seq) data from THP1 (human leukemic monocyte) cells treated with 1,25(OH)$_2$D$_3$ for 4 hr (*Seuter et al., 2016*). The transcriptional response in THP1 cells was dramatically different from that seen in the placenta, with only five upregulated and eight downregulated genes in common. Interestingly, despite this large difference, the promoters of both placenta- and THP1-upregulated genes displayed open chromatin in THP1 cells (*Figure 5—figure supplement 1c*), suggesting that other factors determine the specificity of the vitamin D response in these cells. In contrast, H3K4me3 levels in the placenta were higher for promoters of placenta-upregulated genes than for THP1-upregulated ones (*Figure 5—figure supplement 1d*), suggesting that the 25(OH)D$_3$ response in the placenta is partially guided by the chromatin status. Importantly, these results suggest that an altered epigenetic landscape in the placenta may lead to differences in the response to 25(OH)D$_3$ exposure.

Our data suggest that in human placenta 25(OH)D$_3$ is metabolized to the active 1,25(OH)$_2$D$_3$, which can regulate gene expression at several levels in an intracrine fashion within the same tissue. Whether

the observed 25(OH)D$_3$-induced gene expression changes followed through into protein expression within this short time frame or whether post-translational protein modifications could also be observed in 25(OH)D$_3$-treated placental tissue was explored using LC-MS/MS.

## Short-term 25(OH)D$_3$ exposure leads to proteomic changes in human placenta

The quantitative proteomic analysis of villous fragments from four fresh term human placentas exposed to Tyrode's buffer containing 20 µM 25(OH)D$_3$ plus 0.7 mM BSA (treatment group) compared to samples from the same placenta incubated with Tyrode's buffer containing 0.7 mM BSA (control group) for 8 hr profiled 9279 unique protein groups (peptide level FDR, p<0.05). Of these proteins, 246 were differentially expressed following 8 hr exposure to 25(OH)D$_3$ compared with control (paired permutation testing, p<0.05). 25(OH)D$_3$ incubation resulted in differential expression of 98 methylated, 67 phosphorylated, and 9 acetylated proteins.

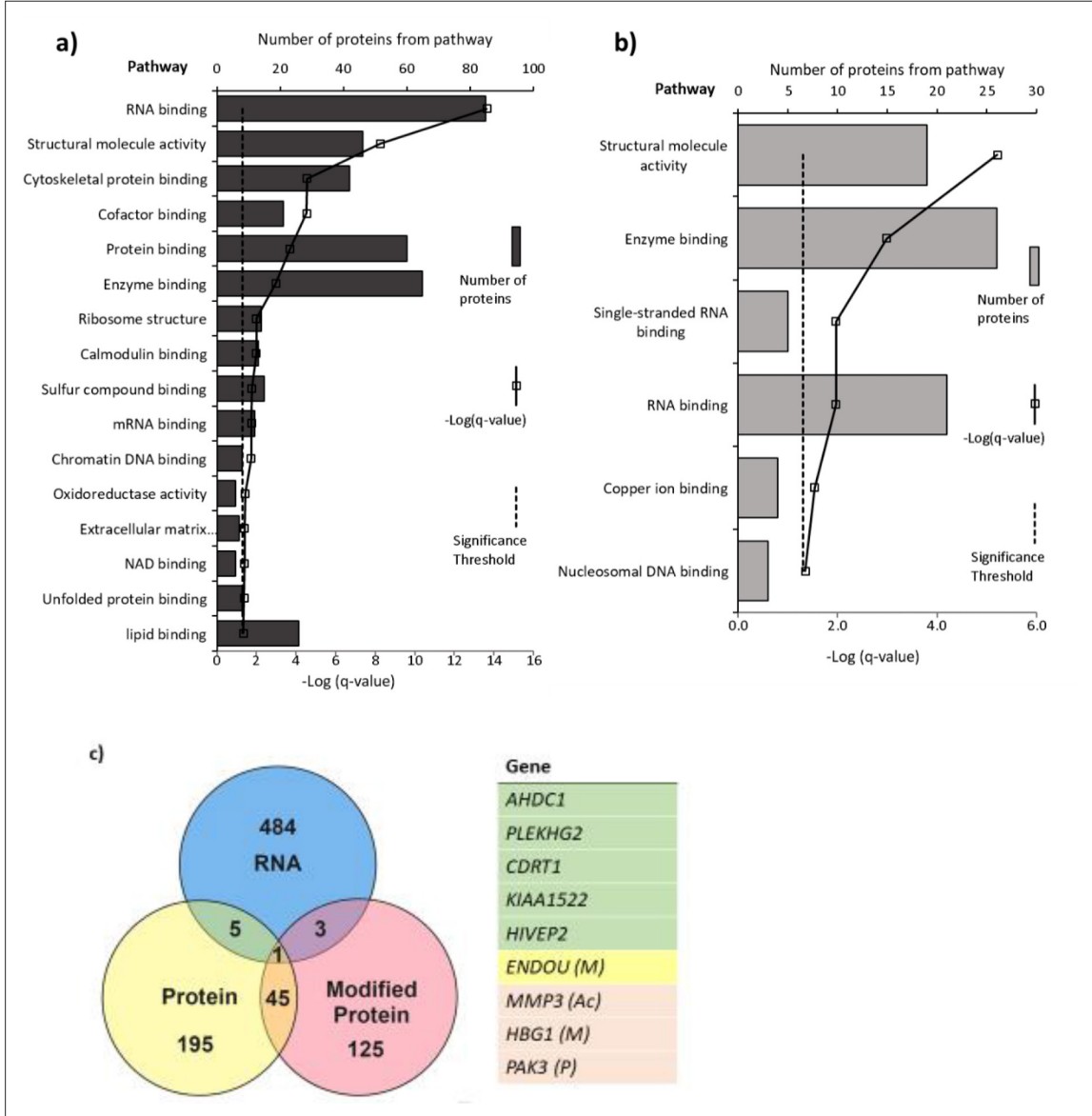

**Figure 6.** Pathways with (**a**) altered protein and (**b**) methylated protein expression in response to 25(OH)D$_3$ treatment. Significantly altered pathways from genes mapped to sites of altered protein expression. Pathways identified using ToppGene and displayed as –log q-value. (**c**) Alignment of RNA and protein expression data. Nine genes were altered at both the RNA and protein level. M, methylated protein; Ac, acetylated protein; P, phosphorylated protein.

A distinct set of proteins were altered in response to 25(OH)D$_3$ in terms of expression level and post-translational modifications. The list of altered mRNA transcripts and altered proteins following 25(OH)D$_3$ treatment was aligned to determine which overlapped (*Figure 6*). Several of the genes shown to be altered at the RNA level could not be measured at the peptide level or did not have changes in protein level within this time frame. 25(OH)D$_3$ altered nine genes at both the gene expression and protein or modified protein level (p<0.05): *HIVEP2* (human immunodeficiency virus type I enhancer binding protein 2), a transcription factor involved in immune function and bone remodeling; *CDRT1* (CMT1A duplicated region transcript 1), an F-box protein that acts as protein-ubiquitin ligase; *KIAA1522*, *AHDC1* (AT-hook DNA binding motif containing 1), involved in DNA binding; *ENDOU* (endonuclease, poly(U) specific) and phosphorylated *PAK3* (P21 [RAC1] activated kinase 3), a serine-threonine kinase that forms an activated complex with GTP-bound RAS-like (P21), CDC2 and RAC1 important for cytoskeletal remodeling. There was decreased gene and protein expression of *PLEKHG2*, methylated *HBG1* (fetal hemoglobin), and acetylated matrix metalloproteinase-3 (*MMP3*), which is involved in the breakdown of extracellular matrix proteins, tissue remodeling, and can also act as a transcription factor.

Pathways significantly enriched in the differentially expressed proteins in 25(OH)D$_3$-treated placenta compared to control include *RNA binding*, *cytoskeletal protein binding*, and *DNA chromatin binding* (e.g., histone proteins) (FDR-corrected p-value<0.05; *Figure 6*). The methylated proteins that were differentially expressed are involved in regulating RNA, DNA, and enzyme binding. One such protein group that showed altered expression was histone proteins and included methylation changes to these proteins (*Figure 6*).

The pathway analysis reveals consistent and convergent patterns of functional gene pathways modified by 25(OH)D$_3$ at both the mRNA and protein level. These included gene pathways relating to structural organization and remodeling, which could enhance placental functions in terms of general implantation and size, increasing the placenta's potential to support fetal development. The effects of 25(OH)D$_3$ also include functional gene pathways relating to the regulation of transcription via DNA binding and epigenetic modifications, supporting the importance of these genes in the placental response to vitamin D. Effects of vitamin D on the machinery that regulates wider gene expression provide a mechanism for vitamin D to have extensive effects on placental function. This supports the actions of vitamin D in terms of increased cellular growth and upregulated functions that would support increased fetal growth. Vitamin D metabolites may therefore regulate gene expression via epigenetic mechanisms or be determined by the underlying epigenetic landscape and chromatin state in a tissue-specific manner.

## Conclusion

This is the first quantitative study demonstrating 25(OH)D$_3$ transfer and metabolism by the human placenta, with widespread effects on the placenta itself. We demonstrate an active 25(OH)D$_3$ uptake mechanism on the maternal-facing side of the placenta and that the placenta influences the levels of 25(OH)D$_3$ and its metabolites 24,25(OH)$_2$D$_3$ and 1,25(OH)$_2$D$_3$ in both the fetal and maternal circulations. Suboptimal placental transport and metabolism of maternal 25(OH)D$_3$ may therefore limit fetal supply and impede fetal development. In support of this, we see associations between placental expression levels of key vitamin D handling and metabolic genes and aspects of fetal size in our cohort study. Placental transfer of 25(OH)D$_3$ metabolites may contribute to the increased maternal 1,25(OH)$_2$D$_3$ concentrations during gestation, which contribute to maternal physiological adaptations that support pregnancy. Effects of vitamin D on fetal development may also be mediated via effects on the placenta itself. Indeed, we show that 25(OH)D$_3$ exposure differentially affects the expression of placental genes and proteins, which together map to multiple gene pathways critical for the placenta's role in pregnancy. Epigenetic analyses suggest that the underlying epigenetic landscape helps to dictate the placental transcriptional response to vitamin D treatment, raising the possibility that environmentally induced epigenetic alterations may reshape the vitamin D response.

These data demonstrate a complex interplay between vitamin D and the placenta to ensure that they can both play optimal roles in supporting fetal growth and maternal adaptations to pregnancy. These data are generated from term placenta and so may not represent differences in function, gene expression, and DNA methylation across gestation, and future studies are needed to determine how our findings relate to earlier stages of gestation (*Simner et al., 2017*). Although we know that key

mediators of this interplay, such as *CYP27B1/CYP24A1/VDR*, have higher expression levels earlier in pregnancy, indicating these effects will be important throughout gestation. This work therefore strongly suggests a role for placental transport and metabolism in mediating the balance of vitamin D distribution and utilization throughout pregnancy. Understanding the regulatory mechanisms and rate-limiting steps in the relationship between vitamin D and the placenta will be a prerequisite for identifying options for targeted intervention to improve pregnancy outcomes.

# Materials and methods

### Key resources table

| Reagent type (species) or resource | Designation | Source or reference | Identifiers | Additional information |
|---|---|---|---|---|
| Sequence-based reagent | CYP24A1_F | This paper | PCR primers | GAAAGAATTGTA TGCTGCTGTCA |
| Sequence-based reagent | CYP24A1_R | This paper | PCR primers | CACATTAGACTG TTTGCTGTCGT |
| Sequence-based reagent | CYP24A1_Probe | Universal ProbeLibrary (human): https://lifescience.roche.com/global_en/brands/universal-probe-library.html | PCR probe | UPL# 78 |
| Sequence-based reagent | CDRT1_F | This paper | PCR primers | TGCAACCCC AAATTACTGCT |
| Sequence-based reagent | CDRT1_R | This paper | PCR primers | GATGTCTTGA TTGAGCCCTGA |
| Sequence-based reagent | CDRT1_Probe | Universal ProbeLibrary | PCR probe | UPL# 74 |
| Sequence-based reagent | CYP27B1_F | This paper | PCR primers | CGCAGCTGT ATGGGGAGA |
| Sequence-based reagent | CYP27B1_R | This paper | PCR primers | CACCTCAAAAT GTGTTAGGATCTG |
| Sequence-based reagent | CYP27B1_Probe | Universal ProbeLibrary | PCR probe | UPL# 53 |
| Sequence-based reagent | HIVEP2_F | This paper | PCR primers | CGGCAAGCT TACATCATCAA |
| Sequence-based reagent | HIVEP2_R | This paper | PCR primers | AGGACGCATC AGGTTTCATC |
| Sequence-based reagent | HIVEP2_Probe | Universal ProbeLibrary | PCR probe | UPL# 38 |
| Sequence-based reagent | PLEKHG2_F | This paper | PCR primers | TCCCCTAGGA TTCTCTGAAGC |
| Sequence-based reagent | PLEKHG2_R | This paper | PCR primers | GGAGGACCCA CACCAAATAA |
| Sequence-based reagent | PLEKHG2_Probe | Universal ProbeLibrary | PCR probe | UPL# 76 |
| Sequence-based reagent | VDR_F | This paper | PCR primers | TCTGTGACCC TAGAGCTGTCC |
| Sequence-based reagent | VDR_R | This paper | PCR primers | TCCTCAGAGGT GAGGTCTCTG |
| Sequence-based reagent | VDR_Probe | Universal ProbeLibrary | PCR probe | UPL# 43 |
| Sequence-based reagent | AHDC1_F | This paper | PCR primers | CCCCAGGACA CCTCTCTACC |

*Continued on next page*

*Continued*

| Reagent type (species) or resource | Designation | Source or reference | Identifiers | Additional information |
|---|---|---|---|---|
| Sequence-based reagent | AHDC1_R | This paper | PCR primers | CATTTAATTCTT CATACCAATCCTTG |
| Sequence-based reagent | AHDC1_Probe | Universal ProbeLibrary | PCR probe | UPL# 38 |
| Sequence-based reagent | CUBN_F | This paper | PCR primers | GGACAATGT CAGAATAG CTTCGT |
| Sequence-based reagent | CUBN_R | This paper | PCR primers | CAGTGGCT AGCAGGGCTTT |
| Sequence-based reagent | CUBN_Probe | Universal ProbeLibrary | PCR probe | UPL# 10 |
| Sequence-based reagent | LRP2_F | This paper | PCR primers | TTGTTTTGAT GCCTCTGATGA |
| Sequence-based reagent | LRP2_R | This paper | PCR primers | AGCTAGGCA TGTTCGCTCAG |
| Sequence-based reagent | LRP2_Probe | Universal ProbeLibrary | PCR probe | UPL# 34 |
| Sequence-based reagent | RXRα_F | This paper | PCR primers | ACATGCAGAT GGACAAGACG |
| Sequence-based reagent | RXRα_R | This paper | PCR primers | TCGAGAGCC CCTTGGAGT |
| Sequence-based reagent | RXRα_Probe | Universal ProbeLibrary | PCR probe | UPL# 26 |

## Placental samples

For placental perfusion, placental fragment culture, and primary term human cytotrophoblast culture experiments, human placentas were collected from term pregnancies immediately after delivery, following written informed consent and with the approval of the South and West Hants Local Research Ethics Committee (11/SC/0529). The study conformed to the Declaration of Helsinki.

To investigate associations between placental gene expression and maternal or fetal characteristics, we used data and samples from the SWS, a cohort study of 3158 live singleton births (*Inskip et al., 2006*). All data were collected with the approval of the South and West Hants Local Research Ethics Committee (276/97, 307/97). Written informed consent was obtained from all participating women and by parents or guardians with parental responsibility on behalf of their children.

## Placental perfusion

Placentas (n = 5) were perfused using the isolated perfused placental cotyledon methodology (*Schneider et al., 1972*), as previously described (*Cleal et al., 2011*). Catheters (Portex, Kent, UK), 15 cm in length, were inserted in the fetoplacental artery (polythene tubing: i.d. 1.0 mm, o.d. 1.6 mm) and fetoplacental vein (PVC tubing: i.d. 2 mm, o.d. 3 mm) of an intact cotyledon and sutured in place. On the maternal side, five 10 cm lengths of polythene tubing (Portex; i.d. 0.58 mm, o.d. 0.96 mm) were inserted through the decidua and into the intervillous space. The fetal circulation and intervillous space were perfused with a modified Earle's bicarbonate buffer (EBB, 1.8 mM $CaCl_2$, 0.4 mM $MgSO_4$, 116.4 mM NaCl, 5.4 mM KCl, 26.2 mM $NaHCO_3$, 0.9 mM $NaH_2PO_4$, 5.5 mM glucose) containing 0.1% BSA, and 5000 IU $l^{-1}$ heparin equilibrated with 95% $O_2$–5% $CO_2$ using roller pumps (Watson Marlow, Falmouth, UK) at 6 and 14 ml $min^{-1}$, respectively. Perfusion of the fetal circulation was established first, and, if fetal venous outflow was >95% of fetal arterial inflow, the maternal-side arterial perfusion with EBB was established 15 min later. Following 45 min of initial perfusion, the maternal arterial perfusion was switched to EBB containing 30 nM $^{13}C$-25(OH)$D_3$ (Sigma-Aldrich, USA). Approximately 1 ml samples of fetoplacental and maternal venous outflow were collected at time points 30, 60, 90, 120,

150, 180, 200, 220, 240, 260, 280, and 300 min. All samples were snap frozen on dry ice and stored at –80°C. At the end of the experiment, the perfused cotyledon was weighed, snap frozen, and stored at –80°C.

## Vitamin D metabolite measures

$^{13}$C-25(OH)D$_3$, $^{13}$C-24,25(OH)$_2$D$_3$, and $^{13}$C-1,25(OH)$_2$D$_3$ were quantified in samples using ultraperformance liquid chromatography hyphenated with electrospray ionization – tandem mass spectrometry (UPLC-MS/MS) (*Assar et al., 2017*). Analysis of extracted, placental vitamin D metabolites was performed on a Waters ACQUITY ultra performance liquid chromatography (UPLC) coupled to a Waters Xevo TQ-XS mass spectrometer (*Li et al., 2019*). 1,25(OH)$_2$D$_3$ in placental perfusate samples was measured using the 1,25-dihydroxy vitamin D EIA kit (Immunodiagnostic Systems, UK).

## Placental fragment culture

Villous tissue fragments of approximately 10 mg were dissected from the placenta and cultured at 37°C in Tyrode's buffer. Following the experiments as described below, buffer was removed, fragments washed in Tyrode's buffer, snap frozen on dry ice, and stored at –80°C.

### Gene expression experiments for effects of vitamin D

Villous fragments were incubated for 8 hr at 37°C in Tyrode's buffer containing 20 µM 25(OH)D$_3$ (Cayman Chemical, MI) with 0.7 mM BSA (Sigma-Aldrich), or control Tyrode's buffer with 0.7 mM BSA (n = 4–15 in triplicate per treatment).

### Gene expression experiments for vitamin D uptake

Villous fragments were incubated for 8 hr at 37°C in Tyrode's buffer containing 20 µM 25(OH)D$_3$ and BSA, plus the endocytic inhibitors 5 mM amiloride hydrochloride hydrate (Sigma-Aldrich), 10 µM cytochalasin D (Sigma-Aldrich), or 2 µM RAP (Innovative Research, USA), which block pinocytosis, classical clathrin-mediated endocytosis, and megalin-mediated endocytosis, respectively (n = 4–5 in triplicate per treatment). Villous fragments had a prior 30 min pre-incubation in the specific blocker at 37°C before the 8 hr incubation. Control villous fragments were incubated in Tyrode's buffer at 37°C for the pre-incubation period and then a further 8 h.

### DNA methylation, RNA-seq, and proteomic experiments

Villous fragments were incubated for 8 hr at 37°C. The treatment group was in Tyrode's buffer containing 20 µM 25(OH)D$_3$ plus 0.7 mM BSA (n = 8 in triplicate). The control group was in Tyrode's buffer with 0.7 mM BSA (n = 8 in triplicate).

### Experiments for microscopy

Villous fragments were incubated with 150 nM FITC-albumin (Sigma-Aldrich) in Tyrode's buffer at 4 and 37°C. Fragments were incubated for 5, 15, or 30 min at each temperature (n = 5 per treatment group). Control fragments were also incubated with 1.43 µM FITC-dextran (Sigma-Aldrich) at 4 and 37°C for 30 min (n = 4 at both temperatures). At each time point, buffer was removed, and villous fragments were fixed in 4% paraformaldehyde and stored at 4°C. After 24 hr, fragments were transferred to 0.1% sodium azide and stored at 4°C. Fragments were further stained for villous stroma by rhodamine-PSA (*Pisum sativum* agglutinin, Vector Laboratories Inc, USA) and the microvillous membrane of the syncytiotrophoblast by biotin-DSL (*Datura stramonium* lectin, Vector Laboratories Inc) and visualized on a SP5 fluorescent confocal microscope. Images were obtained through a series of z-sections through the fragment. The average FITC-protein uptake for each placental fragment was calculated using the software ImageJ (RRID:SCR_003070; https://imagej.net/).

## SWS characterized placentas

Nonpregnant women aged 20–34 years were recruited to the SWS; assessments of maternal anthropometry were performed at study entry and at 11 and 34 weeks' gestation in those who became pregnant. A tape measure was used to measure mid-upper arm circumference from which arm muscle area was derived using the formula (([mid arm circumference – $\pi$ × triceps skinfold thickness)$^2$/4$\pi$] – 6.5) (*Heymsfield et al., 1982*). Measures of fetal size were determined at 19 and 34 weeks' gestation

using a high-resolution ultrasound system (Acuson 128 XP, Aspen and Sequoia Mountain View, CA). Standardized anatomical landmarks were used to measure head circumference, abdominal circumference, crown-rump length, and femur length. Royston's method was used to derive z-scores for ultrasound measurements of size. The method corrects for variation in age at measurement (*Royston, 1995*). At birth, neonatal anthropometric measures were recorded (birth weight, placental weight, head circumference, and crown–heel length). Within 2 weeks of birth, a subset of babies had dual-energy X-ray absorptiometry (DXA) measurements of lean and fat mass using a Lunar DPX instrument with neonatal scan mode and specific pediatric software (pediatric small scan mode, v4.7c, GE Corp., Madison, WI).

Placentas were collected from term SWS pregnancies within 30 min of delivery. Placental weight was measured after removing blood clots, cutting the umbilical cord flush with its insertion into the placenta, trimming away surrounding membranes, and removing the amnion from the basal plate. Five villous tissue samples were selected from each placenta using a stratified random sampling method (to ensure that the selected samples were representative of the placenta as a whole); the maternal decidua was cut off of each sample. Samples were snap frozen in liquid nitrogen and stored at –80°C. For this study, a cohort of 102 placentae was selected from 300 collected in total based on availability of neonatal data. For each placenta, the five samples were pooled and powdered in a frozen tissue press.

## RNA and DNA extraction

RNA was extracted from 30 mg placental samples using the miRNeasy mini kit with the RNase-free DNase Set (QIAGEN, UK) according to the manufacturer's instructions. DNA was extracted from placental fragments using the QIAGWN DNeasy Blood & Tissue Kit (QIAGEN) according to the manufacturer's instructions. RNA and DNA were quantified by UV absorption (NanoDrop 1000, Thermo Scientific, UK). RNA quality was assessed with an RNA 2100 Bioanalyzer (Agilent, USA) and accepted if the RNA integrity number (RIN) was above 6.0.

## Quantitative reverse transcription PCR (qRT-PCR)

Total RNA (0.2 µg) was reverse transcribed into cDNA, and gene expression was measured using qRT-PCR with a Roche Light-Cycler-480. Oligonucleotide probes were supplied by Roche (Human Universal ProbeLibrary [UPL]; Roche, UK) and primers supplied by Eurogentec (Seraing, Belgium). Control genes were selected using the geNorm human Housekeeping Gene Selection Kit (Primer Design Limited, Southampton, UK). For UPL probes, cycle parameters were 95°C for 10 min; 45 cycles of 95°C for 10 s, 60°C for 30 s; then 72°C for 1 s. For geNorm Probes, the cycle parameters were 95°C for 10 min; 50 cycles of 95°C for 15 s, 60°C for 30 s, and 72°C for 15 s. Intra-assay coefficients of variation for each gene were 5–8%. All samples were run on the same plate in triplicate. All SWS placenta mRNA levels are presented relative to the geometric mean of the three human control genes, tyrosine 3-monooxygenase/tryptophan 5-monooxygenase activation protein, zeta polypeptide (*YWHAZ*), ubiquitin C (*UBC*), and topoisomerase (*TOP1*) (*Cleal et al., 2009*). Placental fragment gene expression data was normalized to the geometric mean of *YWHAZ* and *UBC* as *TOP1* expression was altered by vitamin D treatment.

## RNA-sequencing

Placental fragment RNA samples (450 ng) were converted into cDNA libraries using the Illumina TruSeq Stranded mRNA sample preparation kit. Stranded RNA-sequencing was carried out by Expression Analysis (Durham, USA) using HiSeq 2 × 50 bp paired-end sequencing on an Illumina platform. After quality control, reads were mapped to hg38 (Ensembl; March 2017) using HISAT2 v2.0.5 (*Kim et al., 2019*) and counted with HTSeq v0.6.1p1 (*Anders et al., 2015*) using union mode and stranded=reversed settings. Genes were filtered based upon counts per million (CPM), and genes with <1 CPM in at least half of the samples were removed. Samples were normalized using the trimmed mean of M-values (TMM) method, and median plots demonstrated no samples outside 2 standard deviations. Differential expression analysis was carried out using EdgeR (RRID:SCR_012802; http://bioconductor.org/packages/edgeR/; *Robinson et al., 2010*) in the R statistical computing environment (https://www.R-project.org/). Data were filtered to a Benjamini and Hochberg (B&H) FDR-corrected probability of 5%. Genes with altered expression were mapped to pathways using ToppGene Suite

(RRID:SCR_005726; Division of Bioinformatics, Cincinnati Children's Hospital Medical Centre; http://toppgene.cchmc.org/; *Chen et al., 2009*). Pathways were accepted with a minimum hit count of 4 and B&H FDR-corrected q-value of ≤0.05.

## Illumina 850K DNA methylation

DNA methylation analysis was carried out using the Infinium MethylationEPIC array at Barts and the London School of Medicine and Dentistry Genome Centre. Methylation data as β-values were normalized and differentially methylated CpGs were identified by a Wilcoxon signed-rank test with control versus vitamin D treated using a 10% change cutoff. Sites were removed that contain any missing values. All samples met minimal inclusion criteria for analysis as each sample had >75% sites with a detection p-value<$1 \times 10^{-5}$. Probes on X and Y chromosomes were removed. The average CpG methylation levels were calculated for all gene promoters (TSS ± 2 kb).

RNA-sequencing data, ChIP-seq data, and methylation array data that support the findings of this study have been deposited in the Gene Expression Omnibus (GEO; RRID:SCR_005012; https://www.ncbi.nlm.nih.gov/geo/) with the accession code GSE167431.

## Proteomics

### Quantitative proteomics sample processing

Placental villous fragments were snap frozen at –80°C. These were dissolved in 0.5 M triethylammonium bicarbonate, 0.05% sodium dodecyl sulfate, and subjected to pulsed probe sonication (Misonix, Farmingdale, NY). Lysates were centrifuged (16,000 × *g*, 10 min, 4°C) and supernatants measured for protein content using infrared spectroscopy (Merck Millipore, Darmstadt, Germany). Lysates were then reduced, alkylated, and subjected to trypsin proteolysis. Peptides were isobaric stable isotope labeled using the 8-plex iTRAQ reagent kit and analyzed using orthogonal two-dimensional liquid chromatography (offline alkaline C4 reverse phase and online acidic C18 reverse phase) hyphenated with nanospray ionization and high-resolution tandem mass spectrometry, as previously reported by the authors (*Lofthouse et al., 2019*; *Manousopoulou et al., 2018*).

### Database searching

Unprocessed raw files were submitted to Proteome Discoverer 1.4 (RRID:SCR_014477; https://www.thermofisher.com/order/catalog/product/IQLAAEGABSFAKJMAUH) for target decoy searching against the UniProtKB *Homo sapiens* database (https://www.uniprot.org/proteomes/UP000005640), which comprised 20,159 entries (release date January 2015), allowing for up to two missed cleavages, a precursor mass tolerance of 10 ppm, a minimum peptide length of 6, and a maximum of two variable (one equal) modifications of 8-plex iTRAQ (Y), oxidation (M), deamidation (N, Q), or phosphorylation (S, T, Y). Methylthio (C) and iTRAQ (K, Y, and N-terminus) were set as fixed modifications. FDR at the peptide level was set at <0.05. Percent co-isolation excluding peptides from quantitation was set at 50. Reporter ion ratios from unique peptides only were taken into consideration for the quantitation of the respective protein. A permutation test using the normalized iTRAQ ratios of each vitamin D-treated placenta compared to its respective control was performed. Significance was set at p<0.05. In adherence to the Paris Publication Guidelines for the analysis and documentation of peptide and protein identifications (https://doi.org/10.1074/mcp.T400006-MCP200), only proteins identified with at least two unique peptides were further subjected to bioinformatics. All mass spectrometry data have been deposited to the ProteomeXchange Consortium (RRID:SCR_004055; http://www.proteomexchange.org) via the Proteomics Identifications Archive (PRIDE; RRID:SCR_003411; http://www.ebi.ac.uk/pride/) with the dataset identifier PXD011443. Proteins with altered expression were mapped to gene pathways using ToppGene Suite (RRID:SCR_005726). Pathways were accepted with a minimum hit count of 4 and FDR-corrected q-value of ≤0.05.

## Primary term human cytotrophoblast culture

Cytotrophoblast cells were isolated from term human placentas (n = 2) using an adaptation of the method developed by *Kliman et al., 1986*, as described previously (*Desforges et al., 2015*). Isolated cells were plated in culture medium (Dulbecco's modified Eagle's medium and Ham's F-12 1:1, 10% heat-inactivated fetal calf serum, 0.6% glutamine, and the antibiotics 1% gentamicin, 0.2% penicillin, and 0.2% streptomycin) onto 35 mm culture dishes (Nunc) at a density of $2.5 \times 10^6$ and

were maintained in primary culture at 37°C in a humidified incubator (95% air–5% $CO_2$). At 66 hr, matched cell samples had control media or 25(OH)$D_3$ added to a final concentration of 20 μM and were cultured for a period of 24 hr before collection at 90 hr. β-Human chorionic gonadotropin (β-hCG) released by cultured cytotrophoblast cells into the culture media was measured as a marker of syncytialization using a β-hCG ELISA Kit (DRG Diagnostics, Germany). Increasing levels of β-hCG production were observed by 66 hr of culture; average β-hCG levels in the 90 hr culture media were 413.13 mIU ml$^{-1}$. At 90 hr, media were removed, cells were washed in PBS, and fixed in methanol. Fixed cytotrophoblast cells were blocked with BSA and incubated overnight at 4°C with 1/100 mouse anti-human Desmoplakin I + II antibody, (RRID:AB_443375, Abcam ab16434). Cells were washed and incubated for 2 hr at room temperature with 1/250 DAPI (4′,6-diamidine-2′-phenylindole dihydrochloride, Sigma-Aldrich D9542) + 1/500 Alexa Fluor 568 (goat anti-mouse IgG H&L, Abcam ab175473). Cells were again washed and stored under PBS at 4°C until imaging on a Leica SP8 confocal microscope.

## ChIP-seq

Control and vitamin D-treated cytotrophoblast cells were fixed with 1% formaldehyde for 12 min in PBS. After quenching with glycine (final concentration 0.125 M), fixed cells were washed and lysed as previously described (*Latos et al., 2015*). Chromatin was sonicated using a Bioruptor Pico (Diagenode) to an average size of 200–700 bp. Immunoprecipitation was performed using 10 μg of chromatin and 2.5 μg of human antibody (H3K4me3 [RRID:AB_2616052, Diagenode C15410003], H3K4me1 [RRID:AB_306847, Abcam ab8895], and H3K27ac [RRID:AB_2637079, Diagenode C15410196]). Final DNA purification was performed using the GeneJET PCR Purification Kit (Thermo Scientific, K0701) and eluted in 80 μl of elution buffer. ChIP-seq libraries were prepared from 1 to 5 ng eluted DNA using NEBNext Ultra II DNA library Prep Kit (New England Biolabs) with 12 cycles of library amplification. Libraries were sequenced on an Illumina NextSeq 500 with single-ended 75 bp reads at the Barts and the London Genome Centre. Sequencing reads were trimmed with Trim Galore (RRID:SCR_011847), mapped to hg38 genome assembly using bowtie2 (RRID:SCR_016368), and filtered to retain uniquely mapped reads. Peak detection was performed using MACS2 (RRID:SCR_013291) with the `--broad` option. Using DiffBind (RRID:SCR_012918), we detected no peaks with significant differences in enrichment of either histone mark upon 25(OH)$D_3$ treatment. ChIP-seq signal at gene promoters (±1 kb from TSS) was measured as reads per million.

## External data

ENCODE ChIP-seq data from 16-week placentas were extracted as processed peaks (ENCFF707YUS, ENCFF180ADH). RNA-seq data from THP1 cells (Human, RRID:CVCL_0006, GSE69284) were extracted as processed differentially expressed genes. Raw sequencing reads from FAIRE-seq data in THP1 cells (GSE69297) were downloaded and processed as for ChIP-seq data.

## Data analysis

Data are presented as mean (SEM) and analyzed using IBM SPSS Statistics version 20 (RRID:SCR_019096, IBM, Armonk, NY), unless otherwise stated. Data were log transformed if not normally distributed. Significance was set at $p < 0.05$. A two-way ANOVA was used to explore the effects of temperature and time on FITC-albumin uptake in the fragment experiments. qRT-PCR data were analyzed with one-way ANOVA and Tukey's post-hoc test.

For the SWS data, maternal variables that were not normally distributed were transformed logarithmically.

Partial correlations of placental gene expression levels and maternal, fetal, or neonatal variables were calculated, controlling for sex. Neonatal variables were additionally adjusted for gestational age.

Differences in epigenomic data (EPIC arrays, ChIP-seq, FAIRE-seq) at gene promoters were evaluated using nonparametric Wilcoxon tests, with Benjamini–Hochberg correction for multiple comparisons.

## Acknowledgements

CS was funded by a Gerald Kerkut Charitable Trust studentship and BA by Rank Prize and University of Southampton Vice Chancellor's Studentships plus the MRC.

KMG was supported by the UK Medical Research Council (MC_UU_12011/4), the National Institute for Health Research (NIHR Senior Investigator [NF-SI-0515-10042], NIHR Southampton 1000DaysPlus Global Nutrition Research Group [17/63/154], and NIHR Southampton Biomedical Research Centre [IS-BRC-1215-20004]), British Heart Foundation (RG/15/17/3174) and the US National Institute on Aging of the National Institutes of Health (Award No. U24AG047867).

KSJ was supported by the National Institute for Health Research (NIHR) Cambridge Biomedical Research Centre (ISBRC-1215-20014). The NIHR Cambridge Biomedical Research Centre is a partnership between Cambridge University Hospitals NHS Foundation Trust and the University of Cambridge, funded by the NIHR. The views expressed are those of the authors and not necessarily those of the NHS, the NIHR, or the Department of Health and Social Care. Experimental work performed by KSJ and FH at MRC. EWL was supported by Dr Ann Prentice (UK Medical Research Council U105960371).

The SWS has been supported by grants from Medical Research Council (MRC) (4050502589 [MRC LEU]), Dunhill Medical Trust, British Heart Foundation, Food Standards Agency, National Institute for Health Research (NIHR) Southampton Biomedical Research Centre, University of Southampton and University Hospital Southampton NHS Foundation Trust, NIHR Oxford Biomedical Research Centre, University of Oxford, and the European Union's Seventh Framework Programme (FP7/2007-2013), project EarlyNutrition, under grant agreement 289346 and the European Union's Horizon 2020 research and innovation program (LIFECYCLE, grant agreement no. 733206).

EC has been supported by the Wellcome Trust (201268/Z/16/Z) and an NIHR Clinical Lectureship.

Work leading to these results was supported by the BBSRC (HDHL-Biomarkers, BB/P028179/1), as part of the ALPHABET project, supported by an award made through the ERA-Net on Biomarkers for Nutrition and Health (ERA HDHL), Horizon 2020 grant agreement number 696295.

The proteomic analyses (SDG and AM) were financially supported by the National Institutes of Health (R21AI122389) and the Beckman Institute at the California Institute of Technology.

This project has received funding from the European Union's Horizon 2020 research and innovation program under the Marie Skłodowska-Curie grant agreement InvADeRS no. 841172 to JMF.

The electron microscopy image in *Figure 2* was produced with help of the Biomedical imaging unit, Faculty of Medicine, University of Southampton.

## Additional information

### Competing interests

Antigoni Manousopoulou: AM is CSO of Proteas Bioanalytics Inc, BioLabs at the Lundquist Institute. Cory H White: Cory H White is affiliated with Merck Exploratory Science Centre, Merck Research Laboratories. The author has no financial interests to declare. Elizabeth M Curtis: EC reports honoraria/travel support from Eli Lilly, UCB, Pfizer and Amgen outside the submitted work. Martin Hewison: MH has received an honorarium for presenting to Thornton and Ross. Spiros DD Garbis: SG is President and CEO/CTO of Proteas Bioanalytics Inc, BioLabs at the Lundquist Institute. Nicholas C Harvey: NCH reports personal fees, consultancy, lecture fees and honoraria from Alliance for Better Bone Health, AMGEN, MSD, Eli Lilly, Servier, Shire, UCB, Consilient Healthcare, Kyowa Kirin and Internis Pharma, outside the submitted work. The other authors declare that no competing interests exist.

### Funding

| Funder | Grant reference number | Author |
|---|---|---|
| Gerald Kerkut Charitable Trust | | Claire Simner |

| Funder | Grant reference number | Author |
|---|---|---|
| Rank Prize | | Brogan Ashley |
| Medical Research Council | MC UU 12011/4 | Keith M Godfrey |
| National Institute for Health Research | NF-SI-0515-10042 | Keith M Godfrey |
| NIHR Southampton 1000DaysPlus Global Nutrition Research Group | 17/63/154 | Keith M Godfrey |
| NIHR Southampton Biomedical Research Centre | IS-BRC-1215-20004 | Keith M Godfrey |
| British Heart Foundation | RG/15/17/3174 | Keith M Godfrey |
| National Institutes of Health | U24AG047867 | Keith M Godfrey |
| National Institute for Health Research | ISBRC-1215- 20014 | Kerry S Jones |
| Medical Research Council | U105960371 | Kerry S Jones |
| Medical Research Council | 4050502589 | Keith M Godfrey |
| European Union's Seventh Framework Programme | FP7/2007-2013 | Keith M Godfrey |
| European Union's Horizon 2020 research and innovation programme | 733206 | Keith M Godfrey |
| Wellcome Trust | 201268/Z/16/Z | Elizabeth M Curtis |
| NIHR Clinical Lectureship | | Elizabeth M Curtis |
| National Institutes of Health | R21AI122389 | Spiros D Garbis |
| H2020 Marie Skłodowska-Curie Actions | 841172 | Jennifer May Frost |

The funders had no role in study design, data collection and interpretation, or the decision to submit the work for publication.

## Author contributions

Brogan Ashley, Data curation, Formal analysis, Methodology, Validation, Writing – review and editing; Claire Simner, Data curation, Formal analysis, Investigation, Methodology, Writing – review and editing; Antigoni Manousopoulou, Spiros DD Garbis, Data curation, Proteomics Experimental Design, Performed Proteomics analysis and data interpretation, Proteomics Experimental Design, Performed Proteomics analysis and data interpretation, Writing – review and editing; Carl Jenkinson, Metabolism experimental design, analysis and interpretation, Metabolism experimental design, analysis and interpretation, Metabolism experimental design, analysis and interpretation, Metabolism experimental design, analysis and interpretation, Writing – review and editing; Felicity Hey, Kerry S Jones, Martin Hewison, Data curation, Metabolism experimental design, analysis and interpretation, Metabolism experimental design, analysis and interpretation, Metabolism experimental design, analysis and interpretation, Metabolism experimental design, analysis and interpretation, Writing – review and editing; Jennifer M Frost, Data curation, Performed ChIP-seq experiments, Writing – review and editing; Faisal I Rezwan, Cory H White, John W Holloway, Formal analysis, Transcriptomic and DNA methylation analysis and interpretation, Transcriptomic and DNA methylation analysis and interpretation, Transcriptomic and DNA methylation analysis and interpretation, Writing – review and editing; Emma M Lofthouse, Data curation, Methodology, Writing – review and editing; Emily Hyde, Laura DF Cooke, Data curation, Writing – review and editing; Sheila Barton, Sarah R Crozier, Formal analysis, SWS data collection, analysis and interpretation, SWS data collection, analysis and interpretation, SWS data collection, analysis and interpretation, SWS data collection, analysis and interpretation, SWS data collection, analysis and interpretation, SWS data collection, analysis and interpretation, SWS

data collection, analysis and interpretation, SWS data collection, analysis and interpretation, SWS data collection, analysis and interpretation, Writing – review and editing; Pamela Mahon, Elizabeth M Curtis, Rebecca J Moon, Data curation, SWS data collection, analysis and interpretation, SWS data collection, analysis and interpretation, SWS data collection, analysis and interpretation, SWS data collection, analysis and interpretation, SWS data collection, analysis and interpretation, SWS data collection, analysis and interpretation, SWS data collection, analysis and interpretation, SWS data collection, analysis and interpretation, Writing – review and editing; Hazel M Inskip, Data curation, SWS data collection, analysis and interpretation, SWS data collection, analysis and interpretation, Writing – review and editing; Keith M Godfrey, Data curation, SWS data collection, analysis and interpretation, SWS data collection, analysis and interpretation; Cyrus Cooper, Conceptualization, SWS data collection, analysis and interpretation, SWS data collection, analysis and interpretation, SWS data collection, analysis and interpretation, SWS data collection, analysis and interpretation, SWS data collection, analysis and interpretation, SWS data collection, analysis and interpretation, SWS data collection, analysis and interpretation, Writing – review and editing; Rohan M Lewis, Conceptualization, Data curation, Formal analysis, Funding acquisition, Investigation, Methodology, Project administration, Supervision, Writing – review and editing; Miguel R Branco, Conceptualization, Data curation, Formal analysis, Performed bioinformatics analyses, Visualization, Writing – review and editing; Nicholas C Harvey, Conceptualization, Data curation, Funding acquisition, Investigation, Project administration, Resources, SWS data collection, analysis and interpretation, SWS data collection, analysis and interpretation, SWS data collection, analysis and interpretation, SWS data collection, analysis and interpretation, SWS data collection, analysis and interpretation, SWS data collection, analysis and interpretation, SWS data collection, analysis and interpretation, SWS data collection, analysis and interpretation, Supervision, Writing – review and editing; Jane K Cleal, Conceptualization, Data curation, Formal analysis, Funding acquisition, Investigation, Methodology, Project administration, Supervision, Visualization, Writing – original draft, Writing – review and editing

### Author ORCIDs

Carl Jenkinson http://orcid.org/0000-0002-1838-5282
Felicity Hey http://orcid.org/0000-0003-2695-4817
Jennifer M Frost http://orcid.org/0000-0002-3057-6313
Faisal I Rezwan http://orcid.org/0000-0001-9921-222X
Cory H White http://orcid.org/0000-0002-1619-0174
Emma M Lofthouse http://orcid.org/0000-0002-0175-5590
Emily Hyde http://orcid.org/0000-0001-5084-3709
Laura DF Cooke http://orcid.org/0000-0002-8099-9437
Sheila Barton http://orcid.org/0000-0003-4963-4242
Pamela Mahon http://orcid.org/0000-0003-0661-572X
Elizabeth M Curtis http://orcid.org/0000-0002-5147-0550
Rebecca J Moon http://orcid.org/0000-0003-2334-2284
Sarah R Crozier http://orcid.org/0000-0002-9524-1127
Hazel M Inskip http://orcid.org/0000-0001-8897-1749
Keith M Godfrey http://orcid.org/0000-0002-4643-0618
John W Holloway http://orcid.org/0000-0001-9998-0464
Cyrus Cooper http://orcid.org/0000-0003-3510-0709
Kerry S Jones http://orcid.org/0000-0002-7380-9797
Rohan M Lewis http://orcid.org/0000-0003-4044-9104
Martin Hewison http://orcid.org/0000-0001-5806-9690
Spiros DD Garbis http://orcid.org/0000-0002-1050-0805
Miguel R Branco http://orcid.org/0000-0001-9447-1548
Nicholas C Harvey http://orcid.org/0000-0002-8194-2512
Jane K Cleal http://orcid.org/0000-0001-7978-4327

### Ethics

Human subjects: For placental perfusion, placental fragment culture and primary term human cytotrophoblast culture experiments, human placentas were collected from term pregnancies immediately after delivery, following written informed consent and with the approval of the South and West Hants

Local Research Ethics Committee (11/SC/0529). The study conformed to the Declaration of Helsinki. To investigate associations between placental gene expression and maternal or fetal characteristics we used data and samples from the SWS, a cohort study of 3,158 live singleton births [28]. All data were collected with the approval of the South and West Hants Local Research Ethics Committee (276/97, 307/97). Written informed consent was obtained from all participating women and by parents or guardians with parental responsibility on behalf of their children.

## Decision letter and Author response

Decision letter https://doi.org/10.7554/eLife.71094.sa1
Author response https://doi.org/10.7554/eLife.71094.sa2

## Additional files

### Supplementary files
• Supplementary file 1. RNA-seq data.
• Supplementary file 2. Differentially methylated promoters hg19.
• Transparent reporting form

### Data availability

RNA sequencing data, ChIP-seq data and methylation array data that support the findings of this study have been deposited in the Gene Expression Omnibus (GEO) with the accession code GSE167431. All protein mass spectrometry data have been deposited to the ProteomeXchange Consortium via the Proteomics Identifications Archive with the data set identifier PXD011443.

The following datasets were generated:

| Author(s) | Year | Dataset title | Dataset URL | Database and Identifier |
|---|---|---|---|---|
| Ashley B, Simner C, Manousopoulou A, Jenkinson C, Hey F, Frost JM, Rezwan FI, White CH, Lofthouse E, Hyde E, Cooke L, Barton S, Mahon P, Curtis EM, Moon RJ, Crozier SR, Inskip HM, Godfrey KM, Holloway JW, Cooper C, Jones KS, Lewis RH, Hewison M, Garbis SD, Branco MR, Harvey NC, Cleal JK | 2021 | Placental uptake and metabolism as determinants of pregnancy vitamin D status | https://www.ncbi.nlm.nih.gov/geo/query/acc.cgi?acc=GSE167431 | NCBI Gene Expression Omnibus, GSE167431 |
| Ashley B, Simner C, Manousopoulou A, Jenkinson C, Hey F, Frost JM, Rezwan FI, White CH, Lofthouse E, Hyde E, Cooke L, Barton S, Mahon P, Curtis EM, Moon RJ, Crozier SR, Inskip HM, Godfrey KM, Holloway JW, Cooper C, Jones KS, Lewis RH, Hewison M, Garbis SD, Branco MR, Harvey NC, Cleal JK | 2021 | Effects of vitamin D on the placental proteomic profile | https://www.ebi.ac.uk/pride/archive/projects/PXD011443/private | PRIDE, PXD011443 |

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
