## [Editor Report]

The importance of vitamin D and its metabolism to maternal and fetal outcomes is well recognized. Both mother and fetus (in terms of skeletal development and birth weight) are heavily dependent on sufficient 25(OH) vitamin D and its conversion to 1,25(OH)_2_ vitamin D. This interesting work concentrates on the central role of placental function and not maternal physiology to the uptake of 25(OH) vitamin D and its transformation to 1,25(OH)_2_ vitamin D in determining the availability of vitamin D metabolites to the fetus in pregnancy. This work extends substantially our understanding of fundamental human fetal developmental processes.

---

## [Decision Letter]

**Decision letter after peer review:**

Thank you for submitting your article "Placental uptake and metabolism of 25(OH)Vitamin D determines its activity within the fetoplacental unit" for consideration by *eLife*. Your article has been reviewed by 3 peer reviewers, and the evaluation has been overseen by a Reviewing Editor and Mone Zaidi as the Senior Editor. The following individual involves in the review of your submission has agreed to reveal their identity: Bruce Hollis (Reviewer #2).

*Reviewer #1 (Recommendations for the authors):*

1. These studies are well designed, however there are a few areas where the use of controls would strengthen the conclusions and where discussion of the data should be clarified.

2. In Figure 3 it would have been useful to show FITC albumin images that correlate with the experimental paradigm shown in 3D, ie the effect of the blocker on Villus FITC albumin. Additionally, it seems unlikely that blocking three independent methods of uptake (pinocytosis, classical clathrin-dependent endocytosis and megalin-mediated endocytosis) specifically block 25D uptake. How do the authors interpret this data? One would think that if all three pathways were important as suggested by their inhibition of Cyp24a1 expression, that the other two pathways would compensate when the third is blocked. This is not consistent with their claim that megalin plays a major role, based on RAP inhibition of Cyp24a1 expression.

3. The discussion of the studies shown in Figure 5 should be simplified to state that at baseline the methylation of the upregulated and downregulated genes differs, but that vitamin D has no effect nor does vitamin D alter H3K27ac and H3K4 me3. The way it is currently presented suggests that vitamin D does have an effect, which is very misleading.

*Reviewer #2 (Recommendations for the authors):*

The authors have performed an elaborate and detailed study with respect to vitamin D metabolism during pregnancy using placental perfusion technology. This in itself provides a very unique study in vitro. However, the authors need to reconcile an observation from this work with actual human physiology. The authors suggest that placental production of 1,25OHD has a positive effect on the maternal circulating level of that compound. However a study by Greer F et al., J Pediatrics 1984 would suggest otherwise. In that study a pregnancy women with normal placental function but dysfunctional renal 1- hydroxylase activity failed to increase her circulating level of 1,25OHD while receiving 100,000 IU/d vitamin D2 maintenance of serum calcium. Also in that case, fetal levels of 1,25OHD were normal presumably from normal fetal 1-hydroxylase activity. These in vivo observations suggest that placental production of 1,25OHD is not of importance at sustaining maternal levels of that vitamin D metabolite and a normal functional renal hydroxylase would be, a function that this women did not have.*Reviewer #3 (Recommendations for the authors):*

Some data elements require further clarification and should be expounded upon.

1) Further clarification of study results, including validation of transcriptomic and proteomic data with quantitative PCR and Western blotting for some deferentially expressed genes, and the use of VDR knockdown as a control, if possible.

---

## [Author Response]

Reviewer #1 (Recommendations for the authors):1. These studies are well designed, however there are a few areas where the use of controls would strengthen the conclusions and where discussion of the data should be clarified.

We are pleased that the referee agrees that these are well designed studies and have expanded the information with regards to the control groups used; we have clarified the discussion as outlined below.

2. In Figure 3 it would have been useful to show FITC albumin images that correlate with the experimental paradigm shown in 3D, ie the effect of the blocker on Villus FITC albumin.

Unfortunately, FITC-albumin uptake experiments using blockers were not logistically possible. Following our initial FITC-albumin uptake experiments, it was clear that we needed a more direct measure of vitamin D uptake. For this reason, we moved to using the expression of a vitamin D responsive gene (*CYP24A1*) as our output for the subsequent experiments using blockers. This provides evidence that the vitamin D itself had entered the cell and not just the binding protein (albumin).

Additionally, it seems unlikely that blocking three independent methods of uptake (pinocytosis, classical clathrin-dependent endocytosis and megalin-mediated endocytosis) specifically block 25D uptake. How do the authors interpret this data? One would think that if all three pathways were important as suggested by their inhibition of Cyp24a1 expression, that the other two pathways would compensate when the third is blocked. This is not consistent with their claim that megalin plays a major role, based on RAP inhibition of Cyp24a1 expression.

Thank you for raising this point; we did not make it clear that the blockers used in these studies were not specific blockers of discrete uptake mechanisms and therefore do not exactly block three independent mechanisms; instead, they allow us to home-in from a broad mechanism down to a specific one. We have therefore expanded the text to explain this.

The following has been added to line 208 page 7 of the manuscript:

“The application of a range of endocytic inhibitors, which target different aspects of endocytic uptake, aimed to pinpoint the discrete machinery required for the uptake of this particular cargo.”

Amiloride may induce global cellular effects as it inhibits Na^+^/H^+^ exchange, disrupting the actin cytoskeleton (GTPases involved in actin remodeling are highly sensitive to submembranous pH, leading to impaired actin polymerization), which is required for several endocytic pathways involving invagination.

The following has been explained on line 212 page 7:

“Although the Na+/H^+^ exchange inhibitor amiloride was used in our study to inhibit pinocytosis, it has been shown to also inhibit clatherin mediated uptake in epithelial cells [34 Ivanov et al., 2004] and receptor mediated uptake in kidney cells [35 Gekle et al., 2001], suggesting our observed inhibitory effects of amiloride may be due to nonselective blockage of endocytosis via modulation of the actin cytoskeleton.”

This therefore tells us that the vitamin D uptake is via a form of endocytosis or pinocytosis.

Both cytochalasin D and Receptor Associated Protein (RAP) block an aspect of clathrin-mediated endocytosis. Cytochalasin D caps actin filaments inducing their disassembly and inhibits fluid-phase clathrin-mediated uptake. Receptor Associated Protein (RAP), which blocks binding to megalin and cubilin is the most specific blocker and supports a role for megalin-binding mediated uptake (a class of clathrin-mediated uptake).

The following has been explained on line 216 page 7:

“The inhibitory effect of cytochalasin D suggests that 25(OH)D_3_ is taken up by the human placenta using a clathrin-dependent endocytic mechanism, and specifically blocking this mechanism by RAP indicates a role for binding protein mediated uptake via specific megalin receptor binding.”

To make it clear that they are not independent mechanisms the following has been added on line 208 page 7:

“megalin-mediated endocytosis (one type of clathrin-dependent endocytosis).”

It is also stated on line 201 page 7:

“megalin (LDL-related protein 2 (LRP2)) and cubilin (CUBN), which mediate vitamin D internalisation by clathrin-dependant endocytosis.”

3. The discussion of the studies shown in Figure 5 should be simplified to state that at baseline the methylation of the upregulated and downregulated genes differs, but that vitamin D has no effect nor does vitamin D alter H3K27ac and H3K4 me3. The way it is currently presented suggests that vitamin D does have an effect, which is very misleading.

We apologise that this was not clear and have revised the discussion of this figure in the results.

The following has been added on line 292 page 9:

“whilst the methylation levels of the upregulated and downregulated genes identified in the RNA-seq data was not affected by 25(OH)D_3_ treatment, their baseline levels differed”

and on line 305 page 9:

“However, in line with the DNA methylation data, we found that the enrichment of these marks in the upregulated and downregulated genes identified in the RNA-seq data differed at baseline, but was not affected by 25(OH)D_3_ treatment’.”

We have also clarified that both groups were affected on line 308 page 9.

Reviewer #2 (Recommendations for the authors):The authors have performed an elaborate and detailed study with respect to vitamin D metabolism during pregnancy using placental perfusion technology. This in itself provides a very unique study in vitro.

We thank the reviewer for noting the novelty and complexity of this study.

However, the authors need to reconcile an observation from this work with actual human physiology. The authors suggest that placental production of 1,25OHD has an positive effect on the maternal circulating level of that compound. However a study by Greer F et al., J Pediatrics 1984 would suggest otherwise. In that study a pregnancy women with normal placental function but dysfunctional renal 1- hydroxylase activity failed to increase her circulating level of 1,25OHD while receiving 100,000 IU/d vitamin D2 maintenance of serum calcium. Also in that case, fetal levels of 1,25OHD were normal presumably from normal fetal 1-hydroxylase activity. These in vivo observations suggest that placental production of 1,25OHD is not of importance at sustaining maternal levels of that vitamin D metabolite and a normal functional renal hydroxylase would be, a function that this women did not have.

This is an interesting point, and we now refer to this paper in the manuscript discussion (line 151 page 5) and clarify that although the placenta may synthesize 1,25(OH)_2_D, most of this metabolite in the maternal circulation is produced by the maternal kidney (line 146 page 5).

We cannot be certain why this case study by Greer *et al.,* study did not detect higher 1,25(OH)_2_D levels but it is possible that the lower levels of 1,25(OH)_2_D made by the placenta are cleared by the mother or may have been below the level below the level of detection of the assay available in 1984. We also note that measurement techniques for 1,25(OH)_2_D are now generally vastly improved. Consistent with our findings, a second case study of women with renal disease shows that despite impaired renal function there are increased 1,25(OH)_2_D levels during pregnancy, although they are lower than normal (Reference [27] M. Turner, *et al.,* 1988).

The findings of our study demonstrate that the human placenta does produce 1,25(OH)_2_D and releases it into the maternal circulation. The levels of placental 1,25(OH)_2_D secretion observed here would be expected to increase maternal vitamin D levels but would be a modest proportion of overall vitamin D. We have the benefit of using an open loop perfusion system so that we can measure direct release by the placenta before any maternal uptake or metabolism.

Reviewer #3 (Recommendations for the authors):Some data elements require further clarification and should be expounded upon.1) Further clarification of study results, including validation of transcriptomic and proteomic data with quantitative PCR and Western blotting for some deferentially expressed genes, and the use of VDR knockdown as a control, if possible.

We thank the reviewer for these suggestions. For the transcriptomic data, we have demonstrated replication of changes in specific genes in another cohort, as shown in Supplementary Figure 1. To provide validation, for a selection of genes that were either increased, decreased or unchanged in the RNA-seq experiment (Supplementary Figure 1) we have added qPCR measures of mRNA expression change and now refer to this on line 256 page 8.

For the proteomic data, western blotting is a semi-quantitative approach at best, and we do not feel that validation using this is necessary or appropriate. LCMS demonstrates a high level of analysis specificity, sensitivity and selectivity with an analytical confidence above 99% (vs 70-75% of Western blotting, at best).

Furthermore, a strength of this study is that we have measured changes at the gene and protein levels and we can have high confidence where these both change together.

A VDR knockdown would be an interesting control experiment that would demonstrate that vitamin D is acting via the canonical VDR pathway in the placenta rather than a novel non-canonical VDR independent mechanism. While a novel non-canonical pathway for vitamin D signalling would be highly interesting there is no evidence to suggest that it is likely be the case and addressing this question would be a study in itself. Answering this question is not central to the current paper and is not something that can easily be achieved.